# Adaptive Principal Component Regression with Applications to Panel Data

**Anish Agarwal**[*]
Department of IEOR
Columbia University
New York, NY 10027
aa5194@columbia.edu

**Keegan Harris**
School of Computer Science
Carnegie Mellon University
Pittsburgh, PA 15213
keeganh@cs.cmu.edu

**Justin Whitehouse**
School of Computer Science
Carnegie Mellon University
Pittsburgh, PA 15213
jwhiteho@cs.cmu.edu

**Zhiwei Steven Wu**
School of Computer Science
Carnegie Mellon University
Pittsburgh, PA 15213
zhiweiw@cs.cmu.edu

## Abstract

Principal component regression (PCR) is a popular technique for fixed-design *error-in-variables* regression, a generalization of the linear regression setting in which the observed covariates are corrupted with random noise. We provide the first time-uniform finite sample guarantees for online (regularized) PCR whenever data is collected *adaptively*. Since the proof techniques for analyzing PCR in the fixed design setting do not readily extend to the online setting, our results rely on adapting tools from modern martingale concentration to the error-in-variables setting. As an application of our bounds, we provide a framework for experiment design in panel data settings when interventions are assigned adaptively. Our framework may be thought of as a generalization of the synthetic control and synthetic interventions frameworks, where data is collected via an adaptive intervention assignment policy.

## 1 Introduction

An omnipresent task in machine learning, statistics, and econometrics is that of making predictions about outcomes of interest given an action and conditioned on observable covariates. An often overlooked aspect of the prediction task is that in many settings the learner only has access to *imperfect observations* of the covariates, due to e.g. measurement error or inherent randomness in the problem domain. Such settings are sometimes formulated as *error-in-variables* regression: a *learner* is given access to a collection of data $(Z_n, a_n, Y_n)_{n \geq 1}$, where $Z_n \in \mathbb{R}^d$ are the *observed covariates*, $a_n \in \{1, \dots, A\}$ is the *action taken*, and $Y_n \in \mathbb{R}$ is the *outcome* for each observation $n$. Typically, the outcomes are assumed to be generated by a linear model $Y_n := \langle \theta(a_n), X_n \rangle + \xi_n$ and $Z_n := X_n + \epsilon_n$, where $X_n \in \mathbb{R}^d$ are the *true covariates*, $\epsilon_n \in \mathbb{R}^d$ is the covariate noise, $\theta(a_n) \in \mathbb{R}^d$ is an *unknown slope vector* associated with action $a_n$, and $\xi_n \in \mathbb{R}$ is the response noise. Note that the learner does not get to see the true covariates $X_n$. Observe that when $\epsilon_n = 0$ we recover the traditional linear regression setting. Such an error-in-variables model can encompass many forms of data corruption including measurement error, missing values, discretization, and differential privacy—see [10, 5] for details.

Our point of departure from previous work is that we allow the sequence of data $(Z_n, a_n, Y_n)_{n \geq 1}$ to be chosen *adaptively*. In other words, we provide bounds for learning in the error-in-variables regres-

---

[*]For part of this work, Anish was a postdoc at Amazon, Core AI.

37th Conference on Neural Information Processing Systems (NeurIPS 2023).

sion setting when the data seen at the current round $n$ is allowed to depend on the previously-seen data $(Z_m, a_m, Y_m)_{1 \leq m < n}$. Adaptive data collection occurs when the choices of future observations can depend on the inference from previous data, which is common in learning paradigms such as multi-armed bandits [58, 78], active learning [74], and time-series analysis [76, 36]. Similar to prior work on adaptive data collection that shows that valid statistical inference can be done when the true covariates are observed [34, 67, 43, 91, 92], our work provides the first time-uniform finite sample guarantees for error-in-variables regression using adaptively collected data.

Concretely, we focus on analyzing *principal component regression (PCR)* [54], a method that has been shown to be effective for learning from noisy covariates [10, 7] and a central tool for learning from panel data [8, 5, 7]. At a high level, PCR "de-noises" the sequence of observed covariates $(Z_n)_{n \geq 1}$ as $(\widehat{Z}_n)_{n \geq 1}$ by performing hard singular value thresholding, after which a linear model is learned using the observed outcomes $(Y_n)_{n \geq 1}$ and the denoised covariates $(\widehat{Z}_n)_{n \geq 1}$. See Section 3.2 for further technical background on PCR.

## 1.1 Contributions

1. We derive novel time-uniform bounds for an online variant of regularized PCR when the sequence of covariates is chosen adaptively. The techniques used to derive bounds for PCR in the fixed-sample regime [7] do not extend to the setting in which data is collected adaptively; thus, we require new tools and ideas to obtain our results. Specifically, our results rely on applying recent advances in martingale concentration [50, 51], as well as more classical results on self-normalized concentration [3, 32, 33] which are commonly applied to online regression problems, to the error-in-variables setting. As an example of the bounds we obtain, consider the task of estimating the underlying relationship $\theta(a)$ between true (i.e. noiseless) covariates and observations, given access to $n$ adaptively-chosen noisy covariates and their corresponding actions and observations. The $\ell_2$ estimation error of the adaptive PCR estimator $\widehat{\theta}_n(a)$ can be bounded as

$$\|\widehat{\theta}_n(a) - \theta(a)\|_2^2 = \widetilde{O}\left(\frac{1}{\mathrm{snr}_n(a)^2} \kappa(\mathbf{X}_n(a))^2\right)$$

with high probability, where $\mathrm{snr}_n(a)$ is the *signal-to-noise ratio* associated with action $a$ at round $n$ (Definition 3.5), a measure of how well the true covariates stand out from the noise. $\kappa(\mathbf{X}_n(a))$ is the *condition number* of the true covariates. Intuitively, if $\mathrm{snr}_n(a)$ is high the true covariates can be well-separated from the noise, and therefore PCR accurately estimates $\theta(a)$ as long as the true covariates are well-conditioned.

Despite the harder setting we consider, our PCR bounds for adaptively-collected data largely match the bounds currently known for the fixed-sample regime, and even improve upon them in two important ways: (1) Our bounds are *computable*, i.e. they depend on *known* constants and quantities available to the algorithm. (2) Unlike Agarwal et al. [7], our bounds do not depend on the $\ell_1$-norm of $\theta(a)$, i.e., we do not require approximate sparsity of $\theta(a)$ for the bounds to imply consistency. This is important because PCR is a rotationally-invariant algorithm, and so its performance guarantees should not depend on the orientation of the basis representation of the space to be learned. The price we pay for adaptivity is that $\mathrm{snr}_n(a)$ is defined with respect to a bound on the *total* amount of noise seen by the algorithm so far, instead of just the noise associated with the rounds that $a$ is taken. As a result, our bound for $\widehat{\theta}_n(a)$ may not be tight if $a$ is seldomly selected.

2. We apply our PCR results to the problem of online experiment design with *panel data*. In panel data settings, the learner observes repeated, noisy measurements of *units* (e.g. medical patients, subpopulations, geographic locations) under different *interventions* (e.g. medical treatments, discounts, socioeconomic policies) over *time*. This is an ubiquitous method of data collection, and as a result, learning from panel data has been the subject of significant interest in the econometrics and statistics communities (see Section 2). A popular framework for learning from panel data is *synthetic control* (SC) [1, 2], which uses historical panel data to estimate counterfactual unit measurements under control. Synthetic interventions (SI) [8] is a recent generalization of the SC framework which allows for counterfactual estimation under treatment, in addition to control. By leveraging online PCR, we can perform counterfactual estimation of unit-specific treatment effects under both

treatment and control, as in the SI framework. However, unlike the traditional SI framework, we are the first to establish statistical rates for counterfactual unit outcome estimates under different interventions *while allowing for both units and interventions to be chosen adaptively*. Such adaptivity may naturally occur when treatments are prescribed to new units based on the outcomes of previous units. For example, this is the case when the intervention chosen for each unit is the one which appears to be best based on observations in the past.

## 2  Related work

**Error-in-variables regression**    There is a rich literature on error-in-variables regression (e.g. [42, 57, 28, 44, 84, 47, 41]), with research focusing on topics such as high-dimensional [62, 56, 31, 72] and Bayesian settings [70, 81, 40]. Principal component regression (PCR) [54, 22, 10, 7] is a popular method for error-in-variables regression. The results of Agarwal et al. [7] are of particular relevance to us, as they provide finite sample guarantees for the fixed design (i.e. non-adaptive) version of the setting we consider.

**Self-normalized concentration**    There has been a recent uptick in the application of self-normalized, martingale concentration to online learning problems. In short, self-normalized concentration aims to control the growth of processes that have been normalized by a random, or empirical, measure of accumulated variance [32, 33, 50, 51, 88]. Self-normalized concentration has led to breakthroughs in wide-ranging areas of machine learning such as differential privacy [85, 86], PAC-Bayesian learning [30], convex divergence estimation [63], and online learning [87, 29, 3]. Of particular importance for our work are the results of Abbasi-Yadkori et al. [3], which leverage self-normalized concentration results for vector-valued processes [32, 33] to construct confidence ellipsoids for online regression tasks. We take inspiration from these results when constructing our estimation error bounds for PCR in the sequel.

**Learning in panel data settings**    Our application to panel data builds off of the SI framework [8, 9], which itself is a generalization of the canonical SC framework for learning from panel data [1, 2, 52, 35, 17, 60, 89, 13, 14, 59, 16, 20, 23, 27, 39]. In both frameworks, a *latent factor model* is often used to encode structure between units and time-steps [25, 61, 15, 18, 19, 68, 64, 65]. Specifically, it is assumed that unit outcomes are the product of unit- and time/intervention-specific latent factors, which capture the heterogeneity across time-steps, units, and interventions, and allows for the estimation of unit-specific counterfactuals under treatment and control. Other extensions of the SI framework include applications in biology [79], network effects [11], combinatorially-many interventions [4], and intervening under incentives [45, 66]. Finally, there is a growing line of work at the intersection of online learning and panel data. Chen [26] views the problem of SC as an instance of online linear regression, which allows them to apply the regret guarantees of the online learning algorithm *follow-the-leader* [55] to show that the predictions of SC are comparable to those of the best-in-hindsight weighted average of control unit outcomes. Farias et al. [38] build on the SC framework to estimate treatment effects in adaptive experimental design, while minimizing the regret associated with experimentation. The results of Farias et al. [38] are part of a growing line of work on counterfactual estimation and experimental design using multi-armed bandits [69, 77, 90, 24].

## 3  Setting and background

**Notation**    We use boldface symbols to represent matrices. For $N \in \mathbb{N}$, we use the shorthand $[N] := \{1, \ldots, N\}$. Unless specified otherwise, $\|v\|$ denotes the $\ell_2$-norm of a vector $v$, and $\|\mathbf{A}\|_{op}$ the operator norm of matrix $\mathbf{A}$. We use $\mathrm{diag}(a_1, \ldots, a_k)$ to represent a $k \times k$ diagonal matrix with entries $a_1, \ldots, a_k$. For two numbers $a, b \in \mathbb{R}$, we use $a \wedge b$ as shorthand for $\min\{a, b\}$, and $a \vee b$ to mean $\max\{a, b\}$. Finally, $\mathbb{S}^{d-1}$ denotes the $d$-dimensional unit sphere.

### 3.1  Problem setup

We now describe our error-in-variables setting. We consider a *learner* who interacts with an *environment* over a sequence of rounds. At the start of each round $n \geq 1$, the environment generates covariates $X_n \in W^* \subset \mathbb{R}^d$, where $W^*$ is a low-dimensional linear subspace of dimension $\dim(W^*) = r < d$.

We assume that $r$ (but not $W^*$) is known to the learner. Such "low rank" assumptions are reasonable whenever, e.g. data is generated according to a *latent factor model*, a popular assumption in high-dimensional statistical settings [53, 12, 48]. As we will see in Section 5, analogous assumptions are often also made in panel data settings. The learner then observes *noisy* covariates $Z_n := X_n + \epsilon_n$, where $\epsilon_n \in \mathbb{R}^d$ is a random noise vector. Given $Z_n$, the learner selects an *action* $a_n \in [A]$ and observes $Y_n := \langle \theta(a_n), X_n \rangle + \xi_n$, where $\xi_n$ is random noise and $\theta(a)$ for $a \in [A]$ are unknown slope vectors in $W^*$ that parameterize action choices such that $\|\theta(a)\|_2 \leq L$ for some $L \in \mathbb{R}$. We require that the covariate noise is "well-behaved" according to one of the two the following assumptions:

**Assumption 3.1** (**SubGaussian Covariate Noise**). *For any $n \geq 1$, the noise variable $\epsilon_n$ satisfies (a) $\epsilon_n$ is $\sigma$-subGaussian, (b) $\mathbb{E}\epsilon_n = 0$, and (c) $\|\mathbb{E}\epsilon_n\epsilon_n^\top\|_{op} \leq \gamma$, for some constant $\gamma > 0$.*

**Assumption 3.2** (**Bounded Covariate Noise**). *For any $n \geq 1$, the noise variable $\epsilon_n$ satisfies (a) $\|\epsilon_n\| \leq \sqrt{Cd}$, (b) $\mathbb{E}\epsilon_n = 0$, and (c) $\mathbb{E}\epsilon_n\epsilon_n^\top = \Sigma$, for some positive-definite matrix $\Sigma$ satisfying $\|\Sigma\|_{op} \leq \gamma$, for some constant $\gamma > 0$.*

Note that Assumption 3.2 is a special case of Assumption 3.1, which allows us to get stronger results in some settings. We also impose the following constraint on the noise in the outcomes.

**Assumption 3.3** (**SubGaussian Outcome Noise**). *For any $n \geq 1$, the noise variable $\xi_n$ satisfies (a) $\mathbb{E}\xi_n = 0$, (b) $\xi_n$ is $\eta$-subGaussian, and (c) $\mathbb{E}\xi_n^2 \leq \alpha$, for some constant $\alpha$.*

Under this setting, the goal of the learner is to estimate $\theta(a)$ for $a \in [A]$ given an (possibly adaptively-chosen) observed sequence $(Z_n, a_n, Y_n)_{n \geq 1}$. For $n \geq 1$, we define the matrix $\mathbf{Z}_n \in \mathbb{R}^{n \times d}$ to be the matrix of *observed* (i.e. noisy) covariates, with $Z_1, \ldots, Z_n$ as its rows. Similarly, $\mathbf{X}_n = (X_1, \ldots, X_n)^T \in \mathbb{R}^{n \times d}$ is the matrix of *noiseless* covariates (which are unobserved), and $\mathcal{E}_n = (\epsilon_1, \ldots, \epsilon_n)^T \in R^{n \times d}, \mathbf{Y}_n = (Y_1, \ldots, Y_n)^T \in \mathbb{R}^{n \times 1}$, and $\Xi_n = (\xi_1, \ldots, \xi_n)^T \in \mathbb{R}^{n \times 1}$ are defined analogously. For any action $a \in [A]$, let $N_n(a) := \{s \leq n : a_s = n\}$ be the *set of rounds* up to and including round $n$ on which action $a$ was chosen. Likewise, let $c_n(a) := |N_n(a)|$ denote the *number of rounds* by round $n$ on which action $a$ was chosen. For $a \in [A]$, we enumerate $N_n(a)$ in increasing order as $i_1 \leq \cdots \leq i_{c_n(a)}$. Finally, we define $\mathbf{Z}_n(a) \in \mathbb{R}^{c_n(a) \times d}$ to be $\mathbf{Z}(a) = (Z_{i_1}, \ldots, Z_{i_{c_n(a)}})^T$, and define $\mathbf{X}_n(a), \mathcal{E}_n(a), \mathbf{Y}_n(a)$, and $\Xi_n(a)$ analogously.

### 3.2 Principal component regression

**Background on singular value decomposition** Any matrix $\mathbf{A} \in \mathbb{R}^{n \times d}$ may be written in terms of its singular value decomposition $\mathbf{A} = \mathbf{U}\Sigma\mathbf{V}^T$, where $\mathbf{U} \in \mathbb{R}^{n \times d \wedge n}$ and $\mathbf{V} \in \mathbb{R}^{d \times d \wedge n}$ are matrices with orthonormal columns, and $\Sigma = \text{diag}(\sigma_1(\mathbf{A}), \ldots, \sigma_{d \wedge n}(\mathbf{A})) \in \mathbb{R}^{(d \wedge n) \times (d \wedge n)}$ is a diagonal matrix containing the singular values of $\mathbf{A}$, where we assume $\sigma_1(\mathbf{A}) \geq \cdots \geq \sigma_{d \wedge n}(\mathbf{A}) \geq 0$. Given a *truncation level* $k$, we define the truncation of $\mathbf{A}$ onto its top $k$ principal components as $\mathbf{A}_k := \mathbf{U}_k\text{diag}(\sigma_1(\mathbf{A}), \ldots, \sigma_{k \wedge d \wedge n}(\mathbf{A}))\mathbf{V}_k^T$, where $\mathbf{U}_k \in \mathbb{R}^{n \times k \wedge d \wedge n}$ is the matrix with the first $k \wedge d \wedge n$ columns of $\mathbf{U}$, and $\mathbf{V}_k \in \mathbb{R}^{n \times k \wedge d \wedge n}$ is defined analogously. Given such a singular value decomposition, we can define the projection matrix onto the subspace spanned by the top $k$ right singular vectors as $\mathbf{P}_k \in \mathbb{R}^{d \times d}$ given by $\mathbf{P}_k := \mathbf{V}_k\mathbf{V}_k^T$.

For $n \geq 1$, $a \in [A]$, and $\mathbf{Z}_n(a)$ as defined above, we write the $k$-truncated singular value decomposition of $\mathbf{Z}_n(a)$ as $\mathbf{Z}_{n,k}(a) = \widehat{\mathbf{U}}_{n,k}(a)\text{diag}(\sigma_1(\mathbf{Z}_n(a)), \ldots, \sigma_{k \wedge n \wedge d}(\mathbf{Z}_n(a)))\widehat{\mathbf{V}}_{n,k}^T(a)$, and the corresponding projection onto the top $k$ right singular vectors of $\mathbf{Z}_n(a)$ as $\widehat{\mathbf{P}}_{n,k}(a)$. When $k = r$, we leverage the simplified notation $\widehat{\mathbf{P}}_n(a) := \widehat{\mathbf{P}}_{n,r}(a)$. (Recall $r = \dim(W^*)$.) By $\mathbf{P}$, we denote the projection matrix onto the true, underlying subspace $W^*$. While $\mathbf{P}$ is never known, our results leverage the fact that $\widehat{\mathbf{P}}_n(a)$ converges to $\mathbf{P}$ nicely over time. We define the projected noisy covariate matrix matrix to be $\widehat{\mathbf{Z}}_n(a) := \mathbf{Z}_n(a)\widehat{\mathbf{P}}_n(a)$, and define $\widehat{\mathbf{X}}_n(a), \widehat{\mathcal{E}}_n(a)$ similarly. Any quantity with a "$\widecheck{}$" is defined equivalently to quantities with "$\widehat{}$", except with $\mathbf{P}$ in place of $\widehat{\mathbf{P}}_n(a)$. We are now ready to introduce our procedure for estimating $\theta(a)$ for $a \in [A]$, called *adaptive* (or *online*) principal component regression.

**Definition 3.4** (**Adaptive Principal Component Regression**). *Given regularization parameter $\rho \geq 0$ and truncation level $k \in \mathbb{N}$, for $a \in [A]$ and $n \geq 1$ let $\widehat{\mathbf{Z}}_n(a) := \mathbf{Z}_n(a)\widehat{\mathbf{P}}_{n,k}(a)$ and $\widehat{\mathcal{V}}_n(a) := \widehat{\mathbf{Z}}_n(a)^T\widehat{\mathbf{Z}}_n(a) + \rho\widehat{\mathbf{P}}_{n,k}(a)$. Regularized principal component regression estimates $\theta(a)$ as*

$$\widehat{\theta}_n(a) := \widehat{\mathcal{V}}_n(a)^{-1}\widehat{\mathbf{Z}}_n(a)\mathbf{Y}_n(a).$$

Setting $\rho = 0$ recovers the version of PCR used in Agarwal et al. [7]. In words, (unregularized) PCR "denoises" the observed covariates by projecting them onto the subspace given by their $k$-truncation, before estimating $\theta(a)$ via linear regression using the projected covariates. We choose to regularize since it is known that regularization increases the stability of regression-style algorithms. This added stability from regularization allows us to exploit the online regression bounds of Abbasi-Yadkori et al. [3] to measure the performance of our estimates. Throughout the sequel, we only consider adaptive PCR with truncation level $k = r$.

### 3.3 Signal to noise ratio

We now introduce the concept of *signal to noise ratio*, which will be integral in stating and proving our results. The signal to noise ratio provides a measure of how strongly the true covariates (this is the "signal" of the problem, measured through $\sigma_r(\mathbf{X}_n(a))$) stand out sequentially with respect to the "noise" induced by $\mathcal{E}_n$ (which we will measure through the relevant high probability bounds on $\|\mathcal{E}_n\|_{op}$).

**Definition 3.5** (**Signal to Noise Ratio**). *We define the signal to noise ratio associated with an action $a \in [A]$ at round $n$ as*

$$\mathrm{snr}_n(a) := \frac{\sigma_r(\mathbf{X}_n(a))}{U_n},$$

*where $(U_n)_{n \geq 1}$ is a noise-dependent sequence growing as $U_n = O\left(\sqrt{n} + \sqrt{d} + \sqrt{\log\left(\frac{1}{\delta}\log(n)\right)}\right)$.*

The price we pay for adaptivity is encoded directly into the definition of the signal to noise ratio, $\mathrm{snr}_n(a)$. While one may imagine defining $\mathrm{snr}_n(a)$ as the ratio between $\sigma_r(\mathbf{X}_n(a))$ and $\|\mathcal{E}_n(a)\|_{op}$, bounding $\|\mathcal{E}_n(a)\|_{op}$ is a nontrivial task as the rows of $\mathcal{E}_n(a)$ may be strongly correlated. To circumvent this, we apply the trivial bound $\|\mathcal{E}_n(a)\|_{op} \leq \|\mathcal{E}_n\|_{op}$. Thus, *the price of adaptivity in our setting is that the signal from covariates associated with an action $a$ must stand out with respect to the total covariate noise by time $n$*. The growth condition on $U_n$ presented in Definition 3.5 is motivated as follows: w.h.p $\mathbb{E}\|\mathcal{E}_n\|_{op} \approx \sqrt{d} + \sqrt{n}$, and the extra additive $\sqrt{\log\left(\frac{1}{\delta}\log(n)\right)}$ factor is the price we pay for having high probability control of $\|\mathcal{E}_n\|_{op}$ uniformly over rounds. Below we provide an exact definition for $U_n$, as this choice leads to bounds with known constants and simple conditions on $\mathrm{snr}_n(a)$ for validity.

We consider the following two sequences $(U_n)_{n \geq 1}$ in defining signal to noise ratio, which both satisfy $\|\mathcal{E}_n\|_{op} \leq U_n, \forall n \geq 1$ with probability at least $1 - \delta$, per Lemma B.3.

$$U_n^2 := \begin{cases} \beta\left(3\sqrt{n\ell_{\delta/2\mathcal{N}}(n)} + 5\ell_{\delta/2\mathcal{N}}(n)\right) + n\gamma & \text{when Assumption 3.1 holds} \\ \frac{3}{2}\sqrt{nCd\gamma\ell_\delta(n)} + \frac{7}{3}Cd\ell_\delta(n) + n\gamma & \text{when Assumption 3.2 holds.} \end{cases}$$

In the above, $\delta \in (0,1)$, $\ell_\delta(n) := 2\log\log(2n) + \log\left(\frac{d\pi^2}{12\delta}\right)$, $\beta = 32\sigma^2 e^2$, and $\mathcal{N} = 17^d$ is an upper bound on the $1/8$-covering number of $\mathbb{S}^{d-1}$. While the exact forms of the above sequences $(U_n)_{n \geq 1}$ may appear complicated, it is helpful to realize that, under either Assumption 3.1 or 3.2, we have $U_n = O\left(\sqrt{n} + \sqrt{d} + \sqrt{\log\left(\frac{1}{\delta}\log(n)\right)}\right)$, i.e., the growth condition on $U_n$ presented in Definition 3.5 is satisfied.

We can likewise define the *empirical signal to noise ratio associated with action $a$* as $\widehat{\mathrm{snr}}_n(a) := \frac{\sigma_r(\mathbf{Z}_n(a))}{U_n}$. Note that unlike the (true) signal to noise ratio $\mathrm{snr}_n(a)$, the empirical version $\widehat{\mathrm{snr}}_n(a)$ is *computable* by the learner. Thus, it will be integral in stating our empirical-style bounds in the section that follows.

We conclude this section by comparing our notion of signal to noise ratio to that of Agarwal et al. [7], who define $\mathrm{snr}_n(a)$ instead as $\frac{\sigma_r(\mathbf{X}_n(a))}{\sqrt{n} + \sqrt{d}}$, i.e. the ratio of the "signal" in the covariates to the *expected* operator norm of covariate noise $\mathbb{E}\|\mathcal{E}_n\|_{op}$. Since the goal of our work is high-probability (not in-expectation) estimation guarantees for PCR, we believe using high probability bounds on $\|\mathcal{E}_n\|_{op}$ is more natural when defining $\mathrm{snr}_n(a)$.

# 4 Adaptive bounds for online (regularized) PCR

We now present the main results of this work—high-probability, time- and action-uniform bounds measuring the convergence of the PCR estimates $\widehat{\theta}_n(a)$ to the true slope vectors $\theta(a)$. Unlike existing results [6–8], our bounds are valid when the covariates $(X_n)_{n\geq 1}$ and actions $(a_n)_{n\geq 1}$ are determined in an online (potentially adversarial) manner.

We first point out why the analysis of Agarwal et al. [7, 8] breaks down in the setting of adaptive (or online) PCR. First, many of the concentration inequalities leveraged in Agarwal et al. [7] do not hold in the adaptive design setting. As a particular example, the authors leverage the Hanson-Wright inequality [82, 73] for quadratic forms to study how the noisy covariate matrix $\mathbf{Z}_n$ concentrates around the true matrix $\mathbf{X}_n$. This inequality fails to hold when the design points $(X_n)_{n\geq 1}$ depend on the previous observations. Second, the techniques leveraged by Agarwal et al. [8] to extend the convergence guarantees of PCR to the multiple action setting fail to hold when the $n$-th action $a_n$ is selected based on previous observations. Lastly, the bounds presented in [7] are are inherently fixed-time in nature—a simple way to convert existing fixed-time bounds to time-uniform ones would be to perform a union bound over time steps, but that introduces looseness in the bounds.

We are able to construct our bounds by exploiting connections between online PCR and self-normalized martingale concentration [50, 51, 32, 33]. In particular, we combine martingale-based results for constructing confidence ellipsoids for online regression [3, 32, 33] with methods for high-dimensional covariance estimation [83, 80] to prove our results. Exploiting this connection is what allows us to extend the results of Agarwal et al. [7] to the adaptive design, time-uniform setting. We begin with a bound which, up to constants and polylogarthmic factors, captures the rate of convergence of online PCR in terms of (a) the underlying signal to noise ratio and (b) the conditioning of the observed data.

**Theorem 4.1** (**Rate of Convergence for Online PCR**). *Let $\delta \in (0, 1)$ be an arbitrary confidence parameter. Let $\rho > 0$ be chosen to be sufficiently small, as detailed in Appendix F. Further, assume that there is some $n_0 \geq 1$ such that $\mathrm{rank}(\mathbf{X}_{n_0}(a)) = r$ and $\mathrm{snr}_n(a) \geq 2$ for all $n \geq n_0$. Then, with probability at least $1 - O(A\delta)$, simultaneously for all actions $a \in [A]$ and time steps $n \geq n_0$, we have*

$$\|\widehat{\theta}_n(a) - \theta(a)\|_2^2 = \widetilde{O}\left(\frac{1}{\mathrm{snr}_n(a)^2}\kappa(\mathbf{X}_n(a))^2\right),$$

*where $\kappa(\mathbf{X}_n(a)) := \frac{\sigma_1(\mathbf{X}_n(a))}{\sigma_r(\mathbf{X}_n(a))}$ is the condition number (ignoring zero singular values) of $\mathbf{X}_n(a)$.*

Theorem 4.1 is proved in Appendix E. We begin by comparing our bounds to those of Agarwal et al. [7, 8]. At any fixed time, our bounds take on roughly the same form as those of the aforementioned authors, having an inverse quadratic dependence on the signal to noise ratio. To make their bounds non-vacuous, the authors need to make the "soft sparsity" assumption of $\|\theta(a)\|_1 = O(\sqrt{d})$. Our bound, on the other hand, suffers no dependence on the $\ell_1$-norm of the $\theta(a)$'s. This makes intuitive sense, as the specific choice of a basis should not impact the rate of convergence of PCR. However, our bounds pay a price for adaptivity—in particular, the signal to noise ratio associated with an action is defined with respect to a bound on the *total* operator norm of the matrix $\mathcal{E}_n$. If an action is selected very infrequently, the above bound may become loose.

While the above bound is stated in terms of signal to noise ratio, if we make additional assumptions, we can obtain bounds directly in terms of $d, n,$ and $r$. In particular, the following "well-balancing" assumptions suffice.

**Assumption 4.2** (**Well-balancing assumptions**). *For all $n \geq n_0$, the following hold: (a) $\sigma_i(\mathbf{X}_n(a)) = \Theta\left(\sqrt{\frac{c_n(a)d}{r}}\right)$ for all $i \in [r]$, (b) $c_n(a) = \Theta(c_n(a'))$ for all $a, a' \in [A]$, and (c) $A = O(r)$.*

**Corollary 4.3.** *Assume the same setup as Theorem 4.1, and further assume Assumption 4.2 holds. Then with probability at least $1 - O(A\delta)$, simultaneously for all actions $a \in [A]$ and time steps $n \geq n_0$, we have*

$$\|\widehat{\theta}_n(a) - \theta(a)\|_2^2 = \widetilde{O}\left(\frac{r^2}{d \wedge n}\right).$$

Corollary 4.3 shows that Theorem 4.1 obtains the same estimation rate as Theorem 4.1 of Agarwal et al. [7] if assumption Assumption 4.2 holds. This "well-balancing" assumption says roughly that all non-zero singular values of $\mathbf{X}_n$ are of the same order, each action is selected with the same frequency, and that the number of actions is, at most, proportional to dimension of the true, unknown subspace. As noted by Agarwal et al. [7], the assumption of a "well-balanced spectrum" (for $\mathbf{X}_n$) is common in many works in econometrics and robust statistics, and additionally holds with high probability if the entries of $\mathbf{X}_n$ are i.i.d.[62, 21, 37]. Further, it is often the case that there only few available actions (for instance, in the synthetic control literature there are only two actions [2, 1, 38]), justifying the assumption that $A = O(r)$. Lastly, ensuring that each action is played (very roughly) the same number of times can be viewed as a price for adaptivity.

The proof of Theorem 4.1 is immediate as a corollary from the following, more complicated bound. Theorem 4.4 below measures the convergence of $\widehat{\theta}_n(a)$ to $\theta(a)$ in terms of empirical (i.e. observed) quantities. We imagine this bound to be the most practically relevant of our results, as, unlike the results of Agarwal et al. [7], it is directly computable by the learner, involves known constants, and places minimal conditions on the signal to noise ratio.

**Theorem 4.4 (Empirical Guarantees for Online PCR).** *Let $\delta \in (0,1)$ be an arbitrary confidence parameter. Let $\rho > 0$ be chosen to be sufficiently small, as detailed in Appendix F. Further, assume that there is some $n_0 \geq 1$ such that $\mathrm{rank}(\mathbf{X}_{n_0}(a)) = r$ and $\mathrm{snr}_n(a) \geq 2$ for all $n \geq n_0$. Then, with probability at least $1 - O(A\delta)$, simultaneously for all actions $a \in [A]$ and time steps $n \geq n_0$, we have*

$$\left\|\widehat{\theta}_n(a) - \theta(a)\right\|_2^2 \leq \frac{L^2}{\widehat{\mathrm{snr}}_n(a)^2}\left[74 + 216\kappa(\mathbf{Z}_n(a))^2\right] + \frac{2\mathrm{err}_n(a)}{\sigma_r(\mathbf{Z}_n(a))^2},$$

*where $\kappa(\mathbf{Z}_n(a)) := \frac{\sigma_1(\mathbf{Z}_n(a))}{\sigma_r(\mathbf{Z}_n(a))}$, $\|\theta(a)\|_2 \leq L$, and in the above we define the "error" term $\mathrm{err}_n(a)$ to be*

$$\mathrm{err}_n(a) := 32\rho L^2 + 64\eta^2\left(\log\left(\frac{A}{\delta}\right) + r\log\left(1 + \frac{\sigma_1(\mathbf{Z}_n(a))^2}{\rho}\right)\right)$$
$$+ 6\eta^2\sqrt{2c_n(a)\ell_\delta(c_n(a))} + 10\eta^2\ell_\delta(c_n(a)) + 6c_n(a)\alpha.$$

We see that the above bound, with the exception of the third term, more or less resembles the bound presented in Theorem 4.1, just written in terms of the observed covariates $\mathbf{Z}_n(a)$ instead of the true covariates $\mathbf{X}_n(a)$. We view the third term as a slowly growing "error" term. In particular, all terms in the quantity $\mathrm{err}_n(a)$ are either constant, logarithmic in the singular values of $\mathbf{Z}_n(a)$, or linear in $c_n(a)$, the number of times by round $n$ action $a$ has been selected. This implies that $\mathrm{err}_n(a) = \widetilde{O}(n + d)$, ensuring $\mathrm{err}_n(a)$ is dominated by other terms in the asymptotic analysis. We now provide the proof of Theorem 4.4. The key application of self-normalized, martingale concentration comes into play in bounding the quantities that appear in the upper bounds of terms $T_1$ and $T_2$ (to be defined below).

*Proof.* Observe the decomposition, for any $n \geq 1$ and $a \in [A]$
$$\widehat{\theta}_n(a) - \theta(a) = \widehat{\mathbf{P}}_n(a)(\widehat{\theta}_n(a) - \theta(a)) + (\mathbf{P}^\perp - \widehat{\mathbf{P}}_n^\perp(a))\theta(a),$$
where $\mathbf{P}^\perp$ is the projection onto the subspace orthogonal to $W^*$ and $\widehat{\mathbf{P}}_n^\perp(a)$ is the projection onto the subspace orthogonal to the learned subspace (i.e. that spanned by $\mathbf{Z}_{n,r}(a)$). Since $\widehat{\mathbf{P}}_n(a)(\widehat{\theta}_n(a) - \theta(a))$ and $(\mathbf{P}^\perp - \widehat{\mathbf{P}}_n^\perp(a))\theta(a)$ are orthogonal vectors, we have
$$\left\|\widehat{\theta}_n(a) - \theta(a)\right\|_2^2 = \left\|\widehat{\mathbf{P}}_n(a)(\widehat{\theta}_n(a) - \theta(a))\right\|_2^2 + \left\|(\widehat{\mathbf{P}}_n^\perp(a) - \mathbf{P}^\perp)\theta(a)\right\|_2^2.$$
We bound these two terms separately, beginning with the second term. Going forward, fix an action $a \in [A]$. Observe that with probability at least $1 - \delta$, simultaneously for all $n \geq n_0(a)$,

$$\left\|(\widehat{\mathbf{P}}_n^\perp(a) - \mathbf{P}^\perp)\theta(a)\right\|_2^2 \leq \left\|\widehat{\mathbf{P}}_n^\perp(a) - \mathbf{P}^\perp\right\|_{op}^2 \|\theta(a)\|_2^2$$

$$\leq L^2\left\|\widehat{\mathbf{P}}_n^\perp(a) - \mathbf{P}^\perp\right\|_{op}^2 = L^2\left\|\widehat{\mathbf{P}}_n(a) - \mathbf{P}\right\|_{op}^2$$

$$\leq \frac{4L^2U_n^2}{\sigma_r(\mathbf{X}_n(a))^2} \leq \frac{6L^2U_n^2}{\sigma_r(\mathbf{Z}_n(a))^2},$$

where the equality in the above comes from observing $\|\widehat{\mathbf{P}}_n^{\perp}(a) - \mathbf{P}^{\perp}\|_{op} = \|\widehat{\mathbf{P}}_n(a) - \mathbf{P}\|_{op}$, the second-to-last last inequality comes from applying Lemma B.4, and the last inequality follows from the second part of Lemma B.6.

We now bound the first term. Observe that we can write

$$
\begin{aligned}
\left\|\widehat{\mathbf{P}}_n(a)\left(\widehat{\theta}_n(a) - \theta(a)\right)\right\|_2^2 &\leq \frac{1}{\sigma_r(\mathbf{Z}_n(a))^2}\left\|\widehat{\mathbf{Z}}_n(a)\left(\widehat{\theta}_n(a) - \theta(a)\right)\right\|_2^2 \\
&\leq \frac{2}{\sigma_r(\mathbf{Z}_n(a))^2}\left[\underbrace{\left\|\widehat{\mathbf{Z}}_n(a)\widehat{\theta}_n(a) - \mathbf{X}_n(a)\theta(a)\right\|_2^2}_{T_1} + \underbrace{\left\|\mathbf{X}_n(a)\theta(a) - \widehat{\mathbf{Z}}_n(a)\theta(a)\right\|_2^2}_{T_2}\right],
\end{aligned}
\tag{1}
$$

where the first inequality follows from the fact that $\widehat{\mathbf{P}}_n(a) \preceq \frac{1}{\sigma_r(\widehat{\mathbf{Z}}_n(a))^2}\widehat{\mathbf{Z}}_n(a)^{\top}\widehat{\mathbf{Z}}_n(a)$ and $\sigma_r(\mathbf{Z}_n(a)) = \sigma_r(\widehat{\mathbf{Z}}_n(a))$, and the second inequality comes from applying the parallelogram inequality. First we bound $T_1$. We have, with probability at least $1 - O(\delta)$, simultaneously for all $n \geq n_0(a)$

$$
\begin{aligned}
T_1 &\leq 8\left\|\breve{\mathcal{V}}_n(a)^{1/2}\left(\breve{\theta}_n(a) - \theta(a)\right)\right\|_2^2 + 6\|\Xi_n(a)\|_2^2 + 8\left\|\widehat{\mathbf{Z}}_n(a)\theta(a) - \mathbf{X}_n(a)\theta(a)\right\|_2^2 \\
&\leq 32\rho L^2 + 64\eta^2\left(\log\left(\frac{A}{\delta}\right) + r\log\left(1 + \frac{\sigma_1(\mathbf{Z}_n(a))^2}{\rho}\right)\right) + 16L^2 U_n^2 \\
&\quad + 6\eta^2\sqrt{2c_n(a)\ell_\delta(c_n(a))} + 10\eta^2\ell_\delta(c_n(a)) + 6c_n(a)\alpha + 8T_2,
\end{aligned}
\tag{2}
$$

where the first inequality follows from Lemma D.1, and the second inequality follows from applying Lemmas C.1 and D.3. $\ell_\delta(n) = 2\log\log(2n) + \log\left(\frac{d\pi^2}{12\delta}\right)$, as defined in Lemma A.2. We now bound $T_2$. With probability at least $1 - O(\delta)$ simultaneously for all $n \geq n_0$, we have

$$
\begin{aligned}
T_2 &\leq 2L^2\sigma_1(\mathbf{Z}_n(a))^2\left\|\mathbf{P} - \widehat{\mathbf{P}}_n(a)\right\|_{op}^2 + 2L^2\|\mathcal{E}_n\|_{op}^2 \\
&\leq \frac{8L^2\sigma_1(\mathbf{Z}_n(a))^2 U_n^2}{\sigma_r(\mathbf{X}_n(a))^2} + 2L^2 U_n^2 \\
&\leq \frac{12L^2\sigma_1(\mathbf{Z}_n(a))^2 U_n^2}{\sigma_r(\mathbf{Z}_n(a))^2} + 2L^2 U_n^2.
\end{aligned}
\tag{3}
$$

The first inequality follows from Lemma D.2, the second inequality follows from applying Lemmas B.4 and B.3, and the final inequality follows from Lemma B.6.

Piecing the above inequalities together yields the desired result, which can be checked via the argument at the end of Appendix D. A union bound over actions then yields that the desired inequality holds over all actions $a \in [A]$ with probability at least $1 - O(A\delta)$. $\qquad\square$

## 5   Application to causal inference with panel data

We now apply our bounds for adaptive PCR to online experiment design in the context of panel data. In this setting, the learner is interested in estimating *unit-specific counterfactuals* under different *interventions*, given a sequence of unit *outcomes* (or *measurements*) over *time*. Units can range from medical patients, to subpopulations or geographic locations. Examples of interventions include medical treatments, discounts, and socioeconomic policies. *Synthetic control (SC)* is a popular framework used to estimate counterfactual unit outcomes in panel data settings, had they not been treated (i.e. remained under *control*) [1, 2]. In SC, there is a notion of a *pre-intervention* time period in which all units are under control, followed by a *post-intervention* time period, in which every unit undergoes one of several interventions (including control). At a high level, SC fits a model of a unit's pre-treatment outcomes using pre-treatment data from units who remained under control in the post-intervention time period. It then constructs a "synthetic control" by using the learned model to predict the unit's post-intervention outcomes, had they remained under control. *Synthetic interventions (SI)* is a recent generalization of the SC framework, which allows for counterfactual estimation of unit outcomes under different interventions, in addition to control [8]. Using our

bounds from Section 4, we show how to generalize the SI framework of Agarwal et al. [8] to settings where interventions are assigned via an *adaptive intervention assignment policy*.

As a motivating example, consider an online e-commerce platform (learner) which assigns discounts (interventions) to users (units) with the goal of maximizing total user engagement on the platform. For concreteness, suppose that the e-commerce platform assigns discounts *greedily* with respect to the discount level which appears to be best at the current round (i.e. maximizes total engagement for the current user), given the sequence of previously observed (user, discount level, engagement level) tuples. Under such a setting, the intervention assigned at the current round $n$ will be correlated with the observed engagement levels at previous rounds, thus breaking the requirement of the SI framework [8] that the intervention assignment is not adaptive to previously observed outcomes.

Formally, we consider a panel data setting in which the principal observes units over a sequence of rounds. In each round $n$, the learner observes a unit $n$ under *control* for $T_0 \in \mathbb{N}$ time steps, followed by one of $A$ *interventions* (including control, which we denote by 0) for the remaining $T - T_0$ time steps, where $T \in \mathbb{N}$. Overloading notation to be consistent with the literature on panel data, we denote the potential outcome of unit $n$ at time $t$ under intervention $a$ by $Y_{n,t}^{(a)} \in \mathbb{R}$, the set of unit $n$'s pre-treatment outcomes (under control) by $Y_{n,pre} := [Y_{n,1}^{(0)}, \ldots, Y_{n,T_0}^{(0)}]^T \in \mathbb{R}^{T_0}$, and their post-intervention potential outcomes under intervention $a$ by $Y_{n,post}^{(a)} := [Y_{n,T_0+1}^{(a)}, \ldots, Y_{n,T}^{(a)}]^T \in \mathbb{R}^{T-T_0}$. We use $a$ to refer to an arbitrary intervention in $\{0, \ldots, A-1\}$ and $a_n$ to denote the *realized* intervention unit $n$ actually receives in the post-intervention time period. We posit that potential outcomes are generated by the following *latent factor model* over units, time steps, and interventions.

**Assumption 5.1** (**Latent Factor Model**). *Suppose the outcome for unit $n$ at time step $t$ under treatment $a \in \{0, \ldots, A-1\}$ takes the form*

$$Y_{n,t}^{(a)} = \langle U_t^{(a)}, V_n \rangle + \epsilon_{n,t}^{(a)},$$

*where $U_t^{(a)} \in \mathbb{R}^r$ is a latent factor which depends only on the time step $t$ and intervention $a$, $V_n \in \mathbb{R}^r$ is a latent factor which only depends on unit $n$, and $\epsilon_{n,t}^{(a)}$ is zero-mean SubGaussian random noise with variance at most $\sigma^2$. We assume, without loss of generality, that $|\langle U_t^{(a)}, V_n \rangle| \leq 1$ for all $n \geq 1$, $t \in [T]$, $a \in \{0, \ldots, A-1\}$.*

Note that the learner observes $Y_{n,t}^{(a)}$ for only the intervention $a_n$ that unit $n$ is under at time step $t$, and never observes $U_t^{(a)}$, $V_n$, or $\epsilon_{n,t}^{(a)}$. Such "low rank" assumptions are ubiquitous within the panel data literature (see references in Section 2). We assume that $r$ is known to the learner, although principled heuristics exist for estimating $r$ in practice from data (see, e.g. Section 3.2 of Agarwal et al. [8]). Additionally, we make the following "causal transportability" assumption on the latent factors.

**Assumption 5.2** (**Linear span inclusion**). *For any post-intervention time step $t \in [T_0 + 1, T]$ and intervention $a \in \{0, \ldots, A-1\}$, we assume that $U_t^{(a)} \in \text{span}(\{U_t^{(0)} : t \in [T_0]\})$.*

Intuitively, Assumption 5.2 allows for information to be inferred about the potential outcomes in the post-intervention time period using pre-treatment observations. The goal of the learner is to estimate unit-specific counterfactual outcomes under different interventions *when the sequence of units and interventions is chosen adaptively*. In line with previous work in SI and SC, our target causal parameter is the (counterfactual) *average expected post-intervention outcome*.

**Definition 5.3.** *The average expected post-intervention outcome of unit $n$ under intervention $a$ is*

$$\mathbb{E}\bar{Y}_{n,post}^{(a)} := \frac{1}{T - T_0} \sum_{t=T_0+1}^{T} \mathbb{E}Y_{n,t}^{(a)},$$

*where the expectation is taken with respect to $(\epsilon_{n,t}^{(a)})_{T_0 < t \leq T}$.*

While we consider the *average* post-intervention outcome, our results may be readily extended to settings in which the target causal parameter is any *linear* combination of post-intervention outcomes. Next we show that under Assumption 5.1 and Assumption 5.2, $\mathbb{E}\bar{Y}_{n,post}^{(a)}$ may be written as a linear combination of unit $n$'s *pre*-intervention outcomes. We note that similar observations have previously been made in the panel data literature (e.g. [46]), but we include the following lemma for completeness' sake.

**Lemma 5.4** (**Reformulation of average expected post-intervention outcome**). *Under Assumption 5.1 and Assumption 5.2, there exists slope vector $\theta(a) \in \mathbb{R}^{T_0}$, such that the average expected post-intervention outcome of unit $n$ under intervention $a$ is expressible as*

$$\mathbb{E}\bar{Y}_{n,post}^{(a)} = \frac{1}{T - T_0}\langle\theta(a), \mathbb{E}Y_{n,pre}\rangle.$$

$\theta(a)$ may be interpreted as a unit-independent measure of the causal relationship between pre- and post-intervention outcomes. Using this reformulation, adaptive guarantees for the estimation of causal effects over time may now be obtained by applying our online PCR results of Section 4. Overloading the notation of Section 3, we let $\mathbf{X}_n = (\mathbb{E}Y_{1,pre}, \ldots, \mathbb{E}Y_{n,pre})^T$, $\mathbf{Z}_n = (Y_{1,pre}, \ldots, Y_{n,pre})^T$, $\epsilon_{n,pre} = (\epsilon_{n,1}^{(0)}, \ldots, \epsilon_{n,T_0}^{(0)})^T$, $\mathcal{E}_n = (\epsilon_{1,pre}, \ldots, \epsilon_{n,pre})^T$ $\xi_n = \sum_{t=T_0+1}^{T} \epsilon_{n,t}^{(a_n)}$, $\Xi_n = (\xi_1, \ldots, \xi_n)^T$, and

$$\mathbf{Y}_n = \left(\frac{1}{T-T_0}\sum_{t=1}^{T_0} Y_{1,t}^{(a_1)}, \ldots, \frac{1}{T-T_0}\sum_{t=1}^{T_0} Y_{n,t}^{(a_n)}\right)^T.$$

Finally, we define quantities such as $\mathbf{Z}_n(a)$, $\mathbf{X}_n(a)$, $\mathbf{Y}_n(a)$ analogously to Section 3. We now turn to bounding our primary quantity of interest in the panel data setting: prediction error for the average expected post-intervention outcome.

**Theorem 5.5** (**Prediction error of average expected post-intervention outcome**). *Let $\delta \in (0, 1)$ be an arbitrary confidence parameter and $\rho > 0$ be chosen to be sufficiently small, as detailed in Appendix F. Further, assume that Assumptions 5.1 and 5.2 are satisfied, there is some $n_0 \geq 1$ such that $\operatorname{rank}(\mathbf{X}_{n_0}(a)) = r$, and $\operatorname{snr}_n(a) \geq 2$ for all $n \geq n_0$. If $T_0 \leq \frac{1}{2}T$ and $r \leq \sqrt{T_0 \wedge n}$, then under Assumption 4.2 with probability at least $1 - O(A\delta)$, simultaneously for all interventions $a \in \{0, \ldots, A - 1\}$*

$$|\widehat{\mathbb{E}}\bar{Y}_{n,post}^{(a)} - \mathbb{E}\bar{Y}_{n,post}^{(a)}| = \widetilde{O}\left(\frac{L}{\sqrt{T - T_0}} + \frac{r}{\sqrt{T_0 \wedge n}} + \frac{r(L \vee 1)}{\sqrt{(T - T_0)(T_0 \wedge n)}}\right)$$

*where $\widehat{\mathbb{E}}\bar{Y}_{n,post}^{(a)} := \frac{1}{T - T_0} \cdot \langle\widehat{\theta}_n(a), Y_{n,pre}\rangle$ is the estimated average post-intervention outcome for unit $n$ under intervention $a$.*

A more complicated expression which does not require Assumption 4.2 or $T_0 \leq \frac{1}{2}T$ may be found in Appendix G. Observe that $|\widehat{\mathbb{E}}\bar{Y}_{n,post}^{(a)} - \mathbb{E}\bar{Y}_{n,post}^{(a)}| \to 0$ with high probability as $T, T_0, n \to \infty$.

We conclude this section by comparing our results with those of the (non-adaptive) synthetic interventions framework of Agarwal et al. [8]. Since we are regressing over *time*, our method for estimating $\widehat{\mathbb{E}}\bar{Y}_{n,post}^{(a)}$ is known as a *horizontal* regression method in the panel data literature. This is in contrast to *vertical* regression methods, which regress over *units*. See Shen et al. [75] for more details on the similarities and differences between horizontal and vertical regression methods in panel data settings. While we do not exactly match the bound of Agarwal et al. [8] since their synthetic interventions framework of uses a vertical regression method, the two bounds are similar, with the notable differences being as follows: (1) The bound of [8] contains a "slow" $\widetilde{O}(r^{1/2}T_0^{-1/4})$ term which does not appear in our analysis. (2) The non-adaptive SI bound also contains a term which scales as $\widetilde{O}(\frac{\sqrt{n}}{\sqrt{T-T_0}})$ in the worst case, while our bound has no such dependence. However, this comes at the price of slow rate whenever $L$ is large compared to $\sqrt{T - T_0}$.

# 6 Conclusion

We obtain the first adaptive bounds for principal component regression and apply them to the problem of online experiment design in the context of panel data, where we allow for interventions to be assigned according to an adaptive policy. Exciting directions for future work include applications of our results to domains such as differential privacy, and using our bounds to obtain contextual bandit algorithms (e.g. based on LinUCB [3]) capable of regret minimization when given access to noisy contexts.

# Acknowledgements

KH is supported in part by an NDSEG Fellowship. KH, JW, and ZSW are supported in part by NSF FAI Award #1939606. The authors would like to thank the anonymous NeurIPS reviewers for valuable feedback.

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
