# A Results on martingale concentration

In this section, we discuss the basics of martingale concentration that will be used ubiquitously throughout this work. Recall that a process $(M_n)_{n \geq 0}$ is a martingale with respect to some filtration $(\mathcal{F}_n)_{n \geq 0}$ if (a) $(M_n)_{n \geq 0}$ is adapted to $(\mathcal{F}_n)_{n \geq 0}$, (b) $\mathbb{E}|M_n| < \infty$ for all $n \geq 0$, and (c) $\mathbb{E}(M_{n+1} \mid \mathcal{F}_n) = M_n$ for all $n \geq 0$. We denote the "increments" of a martingale as $\Delta M_n := M_n - M_{n-1}$. Results on martingale concentration yield a means of providing tight, time-uniform concentration for statistical tasks in which data is adaptively collected. Recent advances in martingale concentration have allowed for advances in disparate fields of statistical theory such as PAC-Bayesian learning [30], composition in differential privacy [86, 85], and the estimation of convex statistical divergences [63].

Martingale concentration emerges naturally in our work in two ways. First, martingale concentration plays an integral role in bounding the deviations between $\breve{\theta}_t(a)$, the ridge estimate in the true low-dimensional subspace, and $\theta(a)$. In bounding these deviations, we leverage results on the concentration of self-normalized martingale processes, particularly those of de la Peña et al. [32, 33]. These results have long proven useful for calibrating confidence in online linear regression tasks [3, 29], but to the best of our knowledge, we are the first to couple these results with the low-dimensional estimation innate to PCR.

Second, we leverage martingale methods to control the rate at which PCR's estimate of the projection operator onto the unknown subspace converges. To accomplish this, we couple recent breakthroughs on time-uniform, self-normalized concentration for scalar-valued processes [50, 51] with the covering and matrix-CGF approaches for bounding the error in estimates of covariance matrices [83, 71, 80]. While this aspect of our analysis is, more or less, a straightforward merging of two techniques for concentration of measure, it nonetheless requires care due to the technical nature of the machinery being used.

We start by recounting the time-uniform concentration inequality we leverage for controlling the error in the ridge estimate in the true, low-dimensional subspace. The following result is from Abbasi-Yadkori et al. [3], but is a special case of more general, self-normalized concentration results from de la Peña et al. [32, 33].

**Lemma A.1** (**Method of Mixtures**). *Let $(\mathcal{F}_t)_{t \geq 0}$ be a filtration. Let $S_t = \sum_{s=1}^{t} \epsilon_s X_s$ where $(\epsilon_t)_{t \geq 1}$ is an $(\mathcal{F}_t)_{t \geq 0}$-adapted $\mathbb{R}$-valued process of $\sigma$-subGaussian random variables and $(X_t)_{t \geq 1}$ is an $(\mathcal{F}_t)_{t \geq 1}$-predictable $\mathbb{R}^d$-valued process. Let $\rho > 0$ be arbitrary, and let*

$$\mathcal{V}_t := \sum_{s=1}^{t} X_s X_s^T + \rho I_d.$$

*Let $\delta \in (0, 1)$ be any confidence parameter. Then we have with probability at least $1 - \delta$, simultaneously for all $t \geq 1$,*

$$\left\| \mathcal{V}_t^{-1/2} S_t \right\|_2 \leq \sigma \sqrt{2 \log \left( \frac{1}{\delta} \sqrt{\det(\rho^{-1} \mathcal{V}_t)} \right)}.$$

We make several brief comments about the above lemma. First, note that while the word "martingale" doesn't directly appear, the process $(S_n)_{n \geq 0}$ is, in fact, a martingale with respect to the filtration $(\mathcal{F}_n)_{n \geq 0}$. The concentration inequality follows from "mixing" a family of martingales based on $(S_n)_{n \geq 0}$ with respect to some suitable probability measure. Second, how we leverage Lemma A.1 in conjunction with the low-dimensional structure of the problem comes through the presence of $\det(\rho^{-1} \mathcal{V}_n)$ in the bound. In particular, if the sequence $(X_n)_{n \geq 0}$ lies in some low-dimensional subspace $W$, with $\dim(W) = r$, then $\rho^{-1} \mathcal{V}_n$ will have at most $r$ non-unit eigenvalues, and hence $\log \det(\rho^{-1} \mathcal{V}_n) \approx r \log(\|\rho^{-1} \mathcal{V}_n\|_{op})$. We exploit this idea further in the sequel.

Now, we discuss the scalar-valued martingale concentration results from Howard et al. [50, 51] that will be used in our work. Before this, recall that a random variable $X$ is said to be $\sigma$-subGaussian if $\log \mathbb{E} e^{\lambda X} \leq \frac{\lambda^2 \sigma^2}{2}$ for all $\lambda \in \mathbb{R}$. Likewise, we say $X$ is $(\sigma, c)$-subExponential if $\log \mathbb{E} e^{\lambda X} \leq \frac{\lambda^2 \sigma^2}{2}$ for all $|\lambda| < \frac{1}{c}$, and we say $X$ is $(\sigma, c)$-subGamma if $\log \mathbb{E} e^{\lambda X} \leq \sigma^2 \psi_{G,c}(\lambda)$ for all $|\lambda| < \frac{1}{c}$, where $\psi_{G,c}(\lambda) := \frac{\lambda^2}{2(1 - c\lambda)}$. A particularly useful fact in the sequel is that if $X$ is $(\sigma, c)$-subExponential, it is also $(\sigma, c)$-subGamma [50].

**Lemma A.2.** *Suppose $(X_n)_{n \geq 1}$ is a sequence of independent, $(\sigma, c)$-subGamma random variables. Let $(S_n)_{n \geq 0}$ be given by $S_n := \sum_{m=1}^{n} X_n$, and let $\delta \in (0,1)$ be arbitrary. Then, with probability at least $1 - \delta$, simultaneously for all $n \geq 0$, we have*

$$S_n \leq \frac{3}{2}\sigma\sqrt{n\ell_\delta(n)} + \frac{5}{2}c\ell_\delta(n),$$

*where $\ell_\delta(n) := 2\log\log(2n) + \log\left(\frac{d\pi^2}{12\delta}\right)$.*

Note that we have simplified above bound from Howard et al. [51] in several ways. First, the bound as presented in Howard et al. [51] applies to general classes of processes with potentially correlated increments. We have simplified the bound to the setting of independent, subGamma increments to suit our setting. Further, the original bound has many parameters, each of which can be fine-tuned to fit an application. We have pre-selected parameters so that (a) the bound is legible and (b) constants remain relatively small.

We can couple the above bound with a standard argument for bounding error in covariance estimation [83] to obtain high-probability, time-uniform results. The following result will prove useful in measuring the rapidity at which PCA can learn the true, low-dimensional subspace in which the noiseless contexts and slope vectors lie.

**Lemma A.3.** *Let $(\epsilon_n)_{n \geq 1}$ be a sequence of independent, $\sigma$-subGaussian random vectors in $\mathbb{R}^d$. Then, for any $\delta \in (0,1)$, with probability at least $1 - \delta$, simultaneously for all $n \geq 1$, we have*

$$\left\| \sum_{m=1}^{n} \epsilon_m \epsilon_m^\top - \mathbb{E}\epsilon_m \epsilon_m^\top \right\|_{op} \leq \beta\left(3\sqrt{n\ell_{\delta/2\mathcal{N}}(n)} + 5\ell_{\delta/2\mathcal{N}}(n)\right),$$

*where $\ell_\delta$ is as defined in Lemma A.2, $\beta = 32\sigma^2 e^2$, and $\mathcal{N} = 17^d$ is an upper bound on the $1/8$-covering number of $\mathbb{S}^{d-1}$.*

*Proof.* Define the process $(M_n)_{n \geq 0}$ by $M_n := \sum_{m=1}^{n} \epsilon_m \epsilon_m^\top - \mathbb{E}\epsilon_m \epsilon_m^\top$. Using a standard covering argument (more or less verbatim from Wainwright [83]), we have that, if $K \subset \mathbb{S}^{d-1}$ is a minimal $1/8$-covering of $\mathbb{S}^{d-1}$, then

$$\|M_n\|_{op} \leq 2\max_{\nu \in K}\langle \nu, M_n\nu\rangle \tag{4}$$

It is clear that $(M_n)_{n \geq 0}$ is a Hermitian matrix-valued martingale with respect to the natural filtration $(\mathcal{F}_n)_{n \geq 0}$ given by $\mathcal{F}_n := \sigma(\epsilon_m : m \leq n)$. It is thus straightforward to see that, for any $\nu \in \mathbb{S}^{d-1}$, the process $(M_n^\nu)_{n \geq 0}$ given by $M_n^\nu := \nu^T M_n \nu$ is a real-valued martingale with respect to this same filtration.

Moreover, a standard argument (see the proof of Theorem 6.5 in Wainwright [83]) yields that

$$\log\left[\mathbb{E}e^{\lambda\langle\nu, \Delta M_n\nu\rangle}\right] \leq \frac{\lambda^2\beta^2}{2} \qquad \text{for all } |\lambda| < \frac{1}{\beta}, \tag{5}$$

where $\beta := 32e^2\sigma^2$. In other words, for any $\nu \in \mathbb{S}^{d-1}$, the random variable $\langle\nu, \Delta M_n\nu\rangle$ is $(\beta, \beta)$-sub-Exponential, as outlined above. In particular, per the results of Howard et al. [50, 51], this implies that the random variable $\langle\nu, \Delta M_n\nu\rangle$ is $(\beta, \beta)$-subGamma. Applying Lemma A.2, we have, with probability at least $1 - \delta$, simultaneously for all $n \geq 1$ and $\nu \in K$, that

$$|\langle\nu, M_n\nu\rangle| \leq \frac{3}{2}\beta\sqrt{n\ell_{\delta/2\mathcal{N}}(n)} + \beta\frac{5}{2}\ell_{\delta/2\mathcal{N}}(n).$$

Plugging this into Inequality 4, we have that, with probability at least $1 - \delta$, simultaneously for all $n \geq 1$,

$$\|M_n\|_{op} \leq \beta\left(3\sqrt{n\ell_{\delta/2\mathcal{N}}(n)} + 5\ell_{\delta/2\mathcal{N}}(n)\right).$$

$\square$

In the case that the sequence of independent noise variables $(\epsilon_n)_{n \geq 1}$ satisfies $\mathbb{E}\epsilon_n\epsilon_n^\top = \Sigma$ and $\|\epsilon_n\|_2 \leq \sqrt{B}$ uniformly in $n$, we can obtain significantly tighter bounds (in terms of constants). Of particular import is the following bound from Howard et al. [51], which combines their "stitching approach" to time-uniform concentration with matrix chernoff techniques [50, 80, 83] to obtain time-uniform bounds on covariance estimation.

**Lemma A.4.** *Let $(\epsilon_n)_{n\geq 1}$ be a sequence of mean zero, independent random vectors in $\mathbb{R}^d$ such that, for all $n \geq 1$, we have $\mathbb{E}\epsilon_n\epsilon_n^\top = \Sigma$ and $\|\epsilon_n\|_2 \leq \sqrt{Cd}$ almost surely. Further assume $\|\Sigma\|_{op} \leq \gamma$. Then, for any $\delta \in (0,1)$, we have, with probability at least $1 - \delta$, simultaneously for all $n \geq 1$*

$$\left\| \sum_{m=1}^n \left\{ \epsilon_m\epsilon_m^\top - \Sigma \right\} \right\|_{op} \leq \frac{3}{2}\sqrt{nCd\gamma\ell_\delta(n)} + \frac{7}{3}Cd\ell_\delta(t)$$

*where $\ell_\delta(n) := 2\log\log(2n) + \log\left(\frac{d\pi^2}{12\delta}\right)$.*

# B    Convergence results for PCA and singular values

In this appendix, we discuss time-uniform convergence results for adaptive principal component analysis — one half of the principal component regression (PCR) algorithm. In addition, we analyze how singular values of $\mathbf{Z}_n$, the noisy covariate matrix, cluster around those of $\mathbf{X}_n$, the true covariate matrix. We, in the style of Agarwal et al. [7], reduce our study of these quantities to the study of the operator norm of $\mathcal{E}_n$, which we control using the martingale concentration results outlined in Appendix A.

Before continuing, we enumerate several linear algebraic facts that are useful in bounding deviations in singular values.

**Lemma B.1 (Weyl's Inequality).** *Let $\mathbf{A}, \mathbf{B} \in \mathbb{R}^{t\times n}$. Then, for any $i \in [t \wedge n]$, we have*

$$|\sigma_i(\mathbf{A}) - \sigma_i(\mathbf{B})| \leq \|\mathbf{A} - \mathbf{B}\|_{op}.$$

**Lemma B.2 (Wedin's Lemma).** *Let $\mathbf{A}, \mathbf{B} \in \mathbb{R}^{t\times n}$, and suppose $\mathbf{A}$ and $\mathbf{B}$ have spectral decompositions*

$$\mathbf{A} = \mathbf{U}\Sigma\mathbf{V}^T \qquad and \qquad \mathbf{B} = \widehat{\mathbf{U}}\widehat{\Sigma}\widehat{\mathbf{V}}^T.$$

*Then, for any $r \leq n \wedge d$, we have*

$$\max\left\{ \left\| \mathbf{U}_r\mathbf{U}_r^T - \widehat{\mathbf{U}}_r\widehat{\mathbf{U}}_r^T \right\|_{op}, \left\| \mathbf{V}_r\mathbf{V}_r^T - \widehat{\mathbf{V}}_r\widehat{\mathbf{V}}_r^T \right\|_{op} \right\} \leq \frac{2\|\mathbf{A} - \mathbf{B}\|_{op}}{\sigma_r - \sigma_{r+1}},$$

*where $\sigma_r$ (resp. $\sigma_{r+1}$) is the $r$-th (resp. $r+1$-st) largest singular value of $\mathbf{A}$.*

We now prove a time-uniform, high probability bound on the operator norm of $\mathcal{E}_n$. (Recall $\mathcal{E}_n = (\epsilon_1, \ldots, \epsilon_n)^T$.) In particular, we prove a bound for two settings — a looser bound (in terms of constants) which holds when the noise variables $\epsilon_n$ are assumed to be subGaussian, and a tighter, more practically relevant bound that holds when $\epsilon_n$ are assumed to be bounded.

**Lemma B.3 (Covariance Noise Bound).** *Let $(\epsilon_n)_{n\geq 1}$ be a sequence of independent, mean zero random vectors in $\mathbb{R}^d$ such that $\|\mathbb{E}\epsilon_n\epsilon_n^\top\|_{op} \leq \gamma$ for all $n \geq 1$. Let $\delta \in (0,1)$ be an arbitrary confidence parameter. Then, with probability at least $1 - \delta$, simultaneously for all $n \geq 1$ we have*

$$\|\mathcal{E}_n\|_{op}^2 \leq \begin{cases} \beta\left(3\sqrt{n\ell_{\delta/2\mathcal{N}}(n)} + 5\ell_{\delta/2\mathcal{N}}(n)\right) + n\gamma & \text{when Assumption 3.1 holds} \\ \frac{3}{2}\sqrt{nCd\gamma\ell_\delta(n)} + \frac{7}{3}Cd\ell_\delta(n) + n\gamma & \text{when Assumption 3.2 holds.} \end{cases}$$

*where $\ell_\delta$ is as defined in Lemma A.2, $\beta = 32\sigma^2 e^2$, and $\mathcal{N} = 17^d$ is an upper bound on the $1/8$-covering number of $\mathbb{S}^{d-1}$.*

*Proof.* Observe the following basic chain of reasoning.

$$\|\mathcal{E}_n\|_{op}^2 = \|\mathcal{E}_n^\top \mathcal{E}_n\|_{op} \leq \|\mathcal{E}_n^\top \mathcal{E}_n - \mathbb{E}\mathcal{E}_n^\top \mathcal{E}_n\|_{op} + \|\mathbb{E}\mathcal{E}_n^\top \mathcal{E}_n\|_{op}.$$

Now, both Assumption 3.1 and 3.2 imply that $\|\mathbb{E}[\epsilon_n\epsilon_n^\top]\|_{op} \leq \gamma$ for all $n \geq 1$. Consequently, noting that $\mathcal{E}_n^\top \mathcal{E}_n = \sum_{m=1}^n \epsilon_m\epsilon_m^\top$, we have that

$$\|\mathbb{E}\mathcal{E}_n^\top \mathcal{E}_n\|_{op} \leq \sum_{m=1}^n \|\mathbb{E}\epsilon_m\epsilon_m^\top\|_{op} \leq n\gamma.$$

Likewise, we have, applying Lemma A.3 in the case the $\epsilon_n$ satisfy the subGaussian assumption and Lemma A.4 in the case the noise satisfies the bounded assumption, that, with probability at least $1 - \delta$, simultaneously for all $n \geq 1$,

$$\|\mathcal{E}_n^\top \mathcal{E}_n - \mathbb{E}\mathcal{E}_n^\top \mathcal{E}_n\|_{op} \leq \begin{cases} \beta\left(3\sqrt{n\ell_{\delta/2\mathcal{N}}(n)} + 5\ell_{\delta/2\mathcal{N}}(n)\right) & \text{when Assumption 3.1 holds} \\ \frac{3}{2}\sqrt{nCd\gamma\ell_\delta(n)} + \frac{7}{3}Cd\ell_\delta(n) & \text{when Assumption 3.2 holds.} \end{cases}$$

Adding these two lines of reasoning together yields the desired conclusion.

$\square$

We can now apply Lemma B.3 to bound the rate at which $\widehat{\mathbf{P}}_n$, the projection operator onto the learned subspace at time $n$, converges to the projection operator $\mathbf{P}$ onto the true, low-dimensional subspace.

**Lemma B.4** (**Projection Convergence**). *Let $\widehat{\mathbf{P}}_n$ denote the projection operator onto the learned subspace. Further, let $\mathbf{P}$ denote the projection operator onto the true unknown subspace $W^*$. Assume $\mathrm{rank}(\mathbf{X}_{n_0}) = r$ for some $n_0 \geq 1$. Then, for any $\delta \in (0,1)$, we have with probability at least $1 - \delta$, simultaneously for all $n \geq n_0$,*

$$\|\widehat{\mathbf{P}}_n - \mathbf{P}\|_{op}^2 \leq \begin{cases} \frac{4\beta\left(3\sqrt{n\ell_{\delta/2\mathcal{N}}(n)} + 5\ell_{\delta/2\mathcal{N}}(n)\right) + 4n\gamma}{\sigma_r(\mathbf{X}_n)^2} & \text{when Assumption 3.1 holds} \\ \frac{6\sqrt{nCd\gamma\ell_\delta(n)} + \frac{14}{3}Cd\ell_\delta(n) + 4n\gamma}{\sigma_r(\mathbf{X}_n)^2} & \text{when Assumption 3.2 holds.} \end{cases}$$

*Proof.* We write the singular value decompositions of $\mathbf{Z}_n$ and $\mathbf{X}_n$ respectively as

$$\mathbf{Z}_n = \widehat{\mathbf{U}}_n\widehat{\Sigma}_n\widehat{\mathbf{V}}_n^\top \quad \text{and} \quad \mathbf{X}_n = \breve{\mathbf{U}}_n\breve{\Sigma}_n\breve{\mathbf{V}}_n^\top.$$

Since we have assumed $\mathrm{rank}(\mathbf{X}_{n_0}) = r$ for some $n_0 \geq 1$, we have, for all $n \geq n_0$,

$$\widehat{\mathbf{P}}_n = \widehat{\mathbf{V}}_{n,r}\widehat{\mathbf{V}}_{n,r}^T \quad \text{and} \quad \mathbf{P} = \breve{\mathbf{V}}_{n,r}\breve{\mathbf{V}}_{n,r}^T.$$

Now, applying Lemma B.2 and Lemma B.3 we have that, with probability at least $1 - \delta$, simultaneously for all $n \geq n_0$,

$$\begin{aligned} \left\|\widehat{\mathbf{P}}_n - \mathbf{P}\right\|_{op}^2 &= \left\|\widehat{\mathbf{V}}_{n,r}\widehat{\mathbf{V}}_{n,r}^T - \breve{\mathbf{V}}_{n,r}\breve{\mathbf{V}}_{n,r}^T\right\|_{op}^2 \\ &\leq \frac{4\left\|\mathbf{X}_n - \mathbf{Z}_n\right\|_{op}^2}{\sigma_r(\mathbf{X}_n)^2} \\ &= \frac{4\|\mathcal{E}_n\|_{op}^2}{\sigma_r(\mathbf{X}_n)^2} \\ &\leq \begin{cases} \frac{4\beta\left(3\sqrt{n\ell_{\delta/2\mathcal{N}}(n)} + 5\ell_{\delta/2\mathcal{N}}(n)\right) + 4n\gamma}{\sigma_r(\mathbf{X}_n)^2} & \text{when Assumption 3.1 holds} \\ \frac{6\sqrt{nCd\gamma\ell_\delta(n)} + \frac{14}{3}Cd\ell_\delta(n) + 4n\gamma}{\sigma_r(\mathbf{X}_n)^2} & \text{when Assumption 3.2 holds.} \end{cases} \end{aligned}$$

This proves the desired result.

$\square$

We can obtain the following empirical version of Lemma B.4. The proof of the following is identical— the only difference is that, in the application of Wedin's theorem (Lemma B.2), we put the singular values of $\mathbf{Z}_n$ is the denominator instead of $\mathbf{X}_n$. This inequality is, in practice, more useful in computing confidence bounds, as the singular values of $\mathbf{Z}_n$ are computable, whereas the singular values of $\mathbf{X}_n$ are not.

**Lemma B.5** (**Projection Convergence**). *Let $\widehat{\mathbf{P}}_n$ denote the projection operator onto the learned subspace. Further, let $\mathbf{P}$ denote the projection operator onto the true unknown subspace $W^*$. Assume $\mathrm{rank}(\mathbf{X}_{n_0}) = r$ for some $n_0 \geq 1$. Then, for any $\delta \in (0,1)$, we have with probability at least $1 - \delta$, simultaneously for all $n \geq n_0$,*

$$\|\widehat{\mathbf{P}}_n - \mathbf{P}\|_{op} \leq \begin{cases} \frac{4\beta\left(3\sqrt{n\ell_{\delta/2\mathcal{N}}(n)} + 5\ell_{\delta/2\mathcal{N}}(n)\right) + 4n\gamma}{(\sigma_r(\mathbf{Z}_n) - \sigma_{r+1}(\mathbf{Z}_n))^2} & \text{when Assumption 3.1 holds} \\ \frac{6\sqrt{nCd\gamma\ell_\delta(n)} + \frac{14}{3}Cd\ell_\delta(n) + 2n\gamma}{(\sigma_r(\mathbf{Z}_n) - \sigma_{r+1}(\mathbf{Z}_n))^2} & \text{when Assumption 3.2 holds.} \end{cases}$$

What remains is to provide a concentration inequality bounding the deviations between the singular values of the noisy covariate matrix $\mathbf{Z}_n$ and the true, low rank covariate matrix $\mathbf{X}_n$. We have the following inequality.

**Lemma B.6.** *Let $\delta \in (0,1)$ be arbitrary. Then with probability at least $1 - \delta$ we have, simultaneously for all $i \in [r]$ and $n \geq 1$,*

$$|\sigma_i(\mathbf{Z}_n) - \sigma_i(\mathbf{X}_n)|^2 \leq \begin{cases} \beta \left( 3\sqrt{n\ell_{\delta/2\mathcal{N}}(n)} + 5\ell_{\delta/2\mathcal{N}}(n) \right) + n\gamma & \text{when Assumption 3.1 holds} \\ \frac{3}{2}\sqrt{nCd\gamma\ell_\delta(n)} + \frac{7}{3}Cd\ell_\delta(n) + n\gamma & \text{when Assumption 3.2 holds.} \end{cases}$$

*In addition, on the same probability at least $1 - \delta$ event, for all $n \geq 1$ such that $\mathrm{snr}_n \geq 2$, we have*

$$\frac{\sigma_r(\mathbf{X}_n)}{2} \leq \sigma_r(\mathbf{Z}_n) \leq \frac{3\sigma_r(\mathbf{X}_n)}{2}.$$

*Note that the same holds if $\mathrm{snr}_n$ is replaced with the action-specific signal to noise ratio $\mathrm{snr}_n(a)$.*

*Proof.* By Lemma B.1, we know that for any $i \in [r]$,

$$|\sigma_i(\mathbf{Z}_n) - \sigma_i(\mathbf{X}_n)| \leq \|\mathbf{Z}_n - \mathbf{X}_n\|_{op} = \|\mathcal{E}_n\|_{op}.$$

Thus, applying Lemma B.3 yields the first part of the theorem.

Now, suppose $\mathrm{snr}_n \geq 2$. Let $U_n^2$ denote either the first or second line of the already-shown inequality (depending on whether Assumption 3.1 or Assumption 3.2 holds). Then with probability at least $1 - \delta$,

$$\begin{aligned} \sigma_i(\mathbf{Z}_n) &\geq \sigma_i(\mathbf{X}_n) - U_n \\ &= \sigma_i(\mathbf{X}_n) - \sigma_i(\mathbf{X}_n)\frac{U_n}{\sigma_i(\mathbf{X}_n)} \\ &\geq \sigma_i(\mathbf{X}_n) - \sigma_i(\mathbf{X}_n)\frac{U_n}{\sigma_r(\mathbf{X}_n)} \\ &= \sigma_i(\mathbf{X}_n) - \sigma_i(\mathbf{X}_n)\frac{1}{\mathrm{snr}_n} \\ &\geq \frac{\sigma_i(\mathbf{X}_n)}{2}. \end{aligned}$$

$\square$

## C  Convergence results for regression in the true subspace

In this section, we construct confidence ellipsoids bounding the error between $\breve{\theta}_n(a)$ and $\theta(a)$, where

$$\breve{\theta}_n(a) := \breve{\mathcal{V}}_n(a)^{-1}\breve{\mathbf{Z}}_n(a)^\top \mathbf{Y}_n(a),$$

i.e. $\breve{\theta}_n(a)$ is the estimate of $\theta(a)$ given access to the true, underlying subspace $W^*$. While the learner never has direct access to $W^*$, the quantity $\breve{\theta}_n(a)$ proves useful in bounding $\|\widehat{\theta}_n(a) - \theta(a)\|_2^2$, the $\ell_2$ error between the PCR estimate of $\theta(a)$ and $\theta(a)$ itself. In other words, the bounds proved in this section provide a theoretical tool for understanding the convergence of PCR in adaptive settings. The main lemma in this appendix is the following.

**Lemma C.1** (**Error bound in true subspace**). *Let $\delta \in (0,1)$ be an arbitrary confidence parameter. Then, with probability at least $1 - 2\delta$, simultaneously for all $a \in [A]$ and $n \geq 1$, we have*

$$\left\| \breve{\mathcal{V}}_n(a)^{1/2} \left( \breve{\theta}_n(a) - \theta(a) \right) \right\|_2^2 \leq 4\rho L^2 + 8\eta^2 \left[ \log\left(\frac{A}{\delta}\right) + r\log\left(1 + \frac{\sigma_1(\mathbf{Z}_n(a))^2}{\rho}\right) \right]$$

$$+ 2L^2 \begin{cases} \beta \left( 3\sqrt{n\ell_{\delta/2\mathcal{N}}(n)} + 5\ell_{\delta/2\mathcal{N}}(n) \right) + n\gamma & \text{when Assumption 3.1 holds} \\ \frac{3}{2}\sqrt{nCd\gamma\ell_\delta(n)} + \frac{7}{3}Cd\ell_\delta(n) + n\gamma & \text{when Assumption 3.2 holds.} \end{cases}$$

*where $\beta$, $\ell_\delta$, and $\mathcal{N}$ are as defined in Lemma A.3 and A.4.*

To make the proof of the above lemma more modular, we introduce several lemmas. The first of these lemmas simply provides an alternative, easier-to-bound representation of the difference (or error) vector $\breve{\theta}_n(a) - \theta(a)$

**Lemma C.2.** *For any $n \in [N]$ and $a \in [A]$, we have*

$$\breve{\theta}_n(a) - \theta(a) = \breve{\mathcal{V}}_n(a)^{-1}\breve{\mathbf{Z}}_n(a)^\top \Xi_n(a) + \breve{\mathcal{V}}_n(a)^{-1}\breve{\mathbf{Z}}_n(a)^\top \breve{\mathcal{E}}_n(a)\theta(a) - \rho\breve{\mathcal{V}}_n(a)^{-1}\theta(a).$$

*Proof.* A straightforward computation yields

$$
\begin{aligned}
\breve{\theta}_n(a) - \theta(a) &= \breve{\mathcal{V}}_n(a)^{-1}\breve{\mathbf{Z}}_n(a)^\top \mathbf{Y}_n(a) - \theta(a) \\
&= \breve{\mathcal{V}}_n(a)^{-1}\breve{\mathbf{Z}}_n(a)(\mathbf{X}_n(a)\theta(a) + \Xi_n(a)) - \theta(a) \\
&= \breve{\mathcal{V}}_n(a)^{-1}\breve{\mathbf{Z}}_n(a)\mathbf{X}_n(a)\theta(a) + \breve{\mathcal{V}}_n(a)^{-1}\breve{\mathbf{Z}}_n(a)^\top \Xi_n(a) - \theta(a) \\
&\quad \pm \breve{\mathcal{V}}_n(a)^{-1}\breve{\mathbf{Z}}_n(a)\breve{\mathcal{E}}_n(a)\theta(a) \pm \rho\breve{\mathcal{V}}_n(a)^{-1}\theta(a) \\
&= \breve{\mathcal{V}}_n(a)^{-1}\left[\breve{\mathbf{Z}}_n(a)^\top(\mathbf{X}_n(a) + \breve{\mathcal{E}}_n(a)) + \rho I_d\right]\theta(a) + \breve{\mathcal{V}}_n(a)^{-1}\breve{\mathbf{Z}}_n(a)^\top \Xi_n(a) - \theta(a) \\
&\quad - \breve{\mathcal{V}}_n(a)^{-1}\breve{\mathbf{Z}}_n(a)\breve{\mathcal{E}}_n(a)\theta(a) - \rho\breve{\mathcal{V}}_n(a)^{-1}\theta(a) \\
&= \breve{\mathcal{V}}_n(a)^{-1}\left[\breve{\mathbf{Z}}_n(a)^\top\breve{\mathbf{Z}}_n(a) + \rho I_d\right]\theta(a) + \breve{\mathcal{V}}_n(a)^{-1}\breve{\mathbf{Z}}_n(a)^\top \Xi_n(a) - \theta(a) \\
&\quad - \breve{\mathcal{V}}_n(a)^{-1}\breve{\mathbf{Z}}_n(a)\breve{\mathcal{E}}_n(a)\theta(a) - \rho\breve{\mathcal{V}}_n(a)^{-1}\theta(a) \\
&= \breve{\mathcal{V}}_n(a)^{-1}\breve{\mathbf{Z}}_n(a)^\top \Xi_n(a) - \breve{\mathcal{V}}_n(a)^{-1}\breve{\mathbf{Z}}_n(a)^\top \breve{\mathcal{E}}_n(a)\theta(a) - \rho\breve{\mathcal{V}}_n(a)^{-1}\theta(a),
\end{aligned}
$$

which proves the desired result. $\qquad\square$

We leverage the following technical lemma in bounding the determinant of $\breve{\mathcal{V}}_n(a)$, the projection of the covariance matrix $\mathcal{V}_n(a)$ onto the true, unknown subspace $W^*$.

**Lemma C.3** (**Determinant-Trace Inequality**). *Let $\mathbf{A} \in \mathbb{R}^{d \times d}$ be a positive-semidefinite matrix of rank $r$. Then, for any $\rho > 0$, we have*

$$\log\det(I_d + \rho^{-1}\mathcal{A}) \leq r \log\left(1 + \frac{\sigma_1(\mathbf{A})}{\rho}\right).$$

*Proof.* We have

$$
\begin{aligned}
\log\det(I_d + \rho^{-1}\mathbf{A}) &= r \log\left[\det(I_d + \rho^{-1}\mathbf{A})^{1/r}\right] \\
&= r \log\prod_{i=1}^{r}\left(1 + \rho^{-1}\sigma_i(\mathbf{A})\right)^{1/r} \\
&\leq r \log\left(\sum_{i=1}^{r}\frac{1 + \rho^{-1}\sigma_i(\mathbf{A})}{r}\right) \\
&\leq r \log\left(1 + \frac{\sigma_1(\mathbf{A})}{\rho}\right).
\end{aligned}
$$

$\square$

**Lemma C.4.** *Let $\mathbf{A} \in \mathbb{R}^{t \times d}$ be a matrix, and let $\rho > 0$ be arbitrary. Suppose $\mathbf{A}$ has $1 \leq k \leq t \wedge d$ non-zero singular values. Then,*

$$\left\|(\mathbf{A}^T\mathbf{A} + \rho I_d)^{-1/2}\mathbf{A}^T\right\|_{op} \leq \frac{\sigma_k(\mathbf{A})}{\sqrt{\sigma_k(\mathbf{A})^2 + \rho}} \leq 1$$

*Proof.* Let us write the singular value decomposition of $\mathbf{A}$ as

$$\mathbf{A} = \mathbf{U}\Sigma\mathbf{V}^T = \mathbf{U}_k\Sigma_k\mathbf{V}_k^T,$$

where the second equality follows from the fact that $\mathbf{A}$ has exactly $k$ non-zero singular values. We have the equality

$$\mathbf{A}^T\mathbf{A} + \rho I_d = \mathbf{V}(\Sigma_k^2 + \rho I_k)\mathbf{V}^T.$$

Thus, we see

$$\left\|(\mathbf{A}^T\mathbf{A} + \rho I_d)^{-1/2}\mathbf{A}^T\right\|_{op} = \left\|\mathbf{V}_k(\Sigma_k^2 + \rho I_k)^{-1/2}\mathbf{V}_k^T\mathbf{V}_k\Sigma_k\mathbf{U}_k^T\right\|_{op}$$

$$= \left\|(\Sigma_k^2 + \rho I_k)^{-1/2}\Sigma_k\right\|_{op}$$

$$= \max_{i=1}^{k}\frac{\sigma_i(\mathbf{A})}{\sqrt{\rho + \sigma_i(\mathbf{A})^2}} \leq \frac{\sigma_k(\mathbf{A})}{\sqrt{\rho + \sigma_k(\mathbf{A})^2}}.$$

$\square$

With the aforementioned technical lemmas, alongside the time-uniform martingale bounds presented in Appendix A and the singular value concentration results proved in Appendix B, we can now prove Lemma C.1.

***Proof of Lemma C.1.*** First, by Lemma C.2, we see that we have, for any $a \in [A]$ and $n \geq 1$,

$$\breve{\theta}_n(a) - \theta(a) = \breve{\mathcal{V}}_n(a)^{-1}\breve{\mathbf{Z}}_n(a)^\top\Xi_n(a) + \breve{\mathcal{V}}_n(a)^{-1}\breve{\mathbf{Z}}_n(a)^\top\breve{\mathcal{E}}_n(a)\theta(a) - \rho\breve{\mathcal{V}}_n(a)^{-1}\theta(a).$$

With this decomposition in hand, we can apply the parallelogram inequality ($\|x-y\|_2^2 + \|x+y\|_2^2 \leq 2\|x\|_2^2 + 2\|y\|_2^2$) twice to see that

$$\left\|\breve{\mathcal{V}}_n(a)^{1/2}\left(\breve{\theta}_n(a) - \theta(a)\right)\right\|_2^2$$

$$\leq 4\left\|\breve{\mathcal{V}}_t(a)^{-1/2}\breve{\mathbf{Z}}_n(a)^\top\Xi_n(a)\right\|_2^2 + 4\rho^2\left\|\breve{\mathcal{V}}_n(a)^{-1/2}\theta(a)\right\|_2^2 + 2\left\|\breve{\mathcal{V}}_n(a)^{-1/2}\breve{\mathbf{Z}}_n(a)^\top\breve{\mathcal{E}}_n(a)\theta(a)\right\|_2^2$$

$$\leq 4\left\|\breve{\mathcal{V}}_n(a)^{-1/2}\breve{\mathbf{Z}}_n(a)^\top\Xi_n(a)\right\|_2^2 + 4\rho L^2 + 2\left\|\breve{\mathcal{V}}_n(a)^{-1/2}\breve{\mathbf{Z}}_n(a)^\top\breve{\mathcal{E}}_n(a)\theta(a)\right\|_2^2.$$

We bound the first and last terms separately. Applying Lemma A.1 for each $a \in [A]$ and taking a union bound over actions yields that, with probability at least $1 - \delta$, simultaneously for all $n \geq 1$ and $a \in [A]$,

$$\left\|\breve{\mathcal{V}}_n(a)^{-1/2}\breve{\mathbf{Z}}_n(a)^\top\Xi_n(a)\right\|_2 \leq \eta\sqrt{2\log\left(\frac{A}{\delta}\sqrt{\det\left(\rho^{-1}\breve{\mathcal{V}}_n(a)\right)}\right)}$$

$$\leq \eta\sqrt{2\left[\log\left(\frac{A}{\delta}\right) + r\log\left(1 + \frac{\sigma_1(\breve{\mathbf{Z}}_n(a))^2}{\rho}\right)\right]}$$

$$\leq \eta\sqrt{2\left[\log\left(\frac{A}{\delta}\right) + r\log\left(1 + \frac{\sigma_1(\widehat{\mathbf{Z}}_n(a))^2}{\rho}\right)\right]},$$

where the second inequality comes from applying Lemma C.3 and the third inequality comes from the fact that $\sigma_i(\widehat{\mathbf{Z}}_n(a)) \geq \sigma_i(\breve{\mathbf{Z}}_n(a))$ for all $i \in [r]$.

Next, we bound the final term in the above expansion. Observe that

$$\left\|\breve{\mathcal{V}}_n(a)^{-1/2}\breve{\mathbf{Z}}_n(a)^\top\breve{\mathcal{E}}_n(a)\theta(a)\right\|_2 \leq \left\|\breve{\mathcal{V}}_n(a)^{-1/2}\breve{\mathbf{Z}}_n(a)^\top\right\|_{op}\left\|\breve{\mathcal{E}}_n(a)\right\|_{op}\|\theta(a)\|_2.$$

First, note that $\|\theta(a)\|_2 \leq L$ by assumption. Next, Lemma C.4 yields

$$\left\|\breve{\mathcal{V}}_n(a)^{-1/2}\breve{Z}_n(a)^\top\right\|_{op} \leq 1.$$

Lastly, note that by Lemma B.3, we have that, with probability at least $1 - \delta$, simultaneously for all $n \geq 1$

$$\|\breve{\mathcal{E}}_n(a)\|_{op}^2 \leq \|\mathcal{E}_n\|^2 \quad \leq \begin{cases} \beta\left(3\sqrt{n\ell_{\delta/2\mathcal{N}}(n)} + 5\ell_{\delta/2\mathcal{N}}(n)\right) + n\gamma & \text{when Assumption 3.1 holds} \\ \frac{3}{2}\sqrt{nCd\gamma\ell_\delta(n)} + \frac{7}{3}Cd\ell_\delta(n) + n\gamma & \text{when Assumption 3.2 holds.} \end{cases}$$

This proves the desired inequality. $\square$

# D   Technical lemmas for Theorem 4.4

In this appendix, we present several additional technical lemmas that are needed for bounding terms in the main theorem.

**Lemma D.1.** *Assuming the setup of Theorem 4.4, we have, for all $n \geq 1$,*

$$\left\|\widehat{\mathbf{Z}}_n(a)\widehat{\theta}_n(a) - \mathbf{X}_n(a)\theta(a)\right\|_2^2 \leq 8\left\|\breve{\mathcal{V}}_n(a)^{1/2}\left(\breve{\theta}_n(a) - \theta(a)\right)\right\|_2^2 + 6\left\|\Xi_n(a)\right\|_2^2 + 8\left\|\widehat{\mathbf{Z}}_n(a)\theta(a) - \mathbf{X}_n(a)\theta(a)\right\|_2^2$$

*Proof.* A relatively straightforward computation yields

$$\begin{aligned}
\left\|\widehat{\mathbf{Z}}_n(a)\widehat{\theta}_n(a) - \mathbf{X}_t(a)\theta(a)\right\|_2^2 &= \left\|\widehat{\mathbf{Z}}_n(a)\widehat{\theta}_n(a) - \mathbf{X}_n(a)\theta(a) + \Xi_n(a) - \Xi_n(a)\right\|_2^2 \\
&\leq 2\left\|\widehat{\mathbf{Z}}_n(a)\widehat{\theta}_n(a) - \mathbf{Y}_n(a)\right\|_2^2 + 2\left\|\Xi_n(a)\right\|_2^2 \\
&\leq 2\left\|\widehat{\mathbf{Z}}_n(a)\breve{\theta}_n(a) - \mathbf{Y}_n(a)\right\|_2^2 + 2\left\|\Xi_n(a)\right\|_2^2 \\
&\leq 4\left\|\widehat{\mathbf{Z}}_n(a)\breve{\theta}_n(a) - \mathbf{X}_n(a)\theta(a)\right\|_2^2 + 6\left\|\Xi_n(a)\right\|_2^2 \\
&\leq 8\left\|\widehat{\mathbf{Z}}_n(a)\left(\breve{\theta}_n(a) - \theta(a)\right)\right\|_2^2 + 6\left\|\Xi_n(a)\right\|_2^2 + 8\left\|\widehat{\mathbf{Z}}_n(a)\theta(a) - \mathbf{X}_n(a)\theta(a)\right\|_2^2 \\
&= 8\left\|\widehat{\mathbf{Z}}_n(a)\mathbf{P}\left(\breve{\theta}_n(a) - \theta(a)\right)\right\|_2^2 + 6\left\|\Xi_n(a)\right\|_2^2 + 8\left\|\widehat{\mathbf{Z}}_n(a)\theta(a) - \mathbf{X}_n(a)\theta(a)\right\|_2^2 \\
&\leq 8\left\|\breve{\mathbf{Z}}_n(a)\left(\breve{\theta}_n(a) - \theta(a)\right)\right\|_2^2 + 6\left\|\Xi_n(a)\right\|_2^2 + 8\left\|\widehat{\mathbf{Z}}_n(a)\theta(a) - \mathbf{X}_n(a)\theta(a)\right\|_2^2 \\
&\leq 8\left\|\breve{\mathcal{V}}_n(a)^{1/2}\left(\breve{\theta}_n(a) - \theta(a)\right)\right\|_2^2 + 6\left\|\Xi_n(a)\right\|_2^2 + 8\left\|\widehat{\mathbf{Z}}_n(a)\theta(a) - \mathbf{X}_n(a)\theta(a)\right\|_2^2
\end{aligned}$$

where we apply the the parallelogram inequality to obtain the first and third inequalities. The second inequality follows from the characterization of ridge regression in Fact F.2 and the fact $\rho$ is chosen sufficiently small. □

**Lemma D.2.** *Assuming the setup of Theorem 4.4, we have, for all $n \geq 1$,*

$$\left\|\mathbf{X}_n(a)\theta(a) - \widehat{\mathbf{Z}}_n(a)\theta(a)\right\|_2^2 \leq 2L^2\sigma_1(\mathbf{Z}_n(a))^2\left\|\mathbf{P} - \widehat{\mathbf{P}}_n(a)\right\|_{op}^2 + 2L^2\left\|\mathcal{E}_n\right\|_{op}^2$$

*Proof.* A relatively simple computation yields

$$\begin{aligned}
\left\|\mathbf{X}_n(a)\theta(a) - \widehat{\mathbf{Z}}_n(a)\theta(a)\right\|_2^2 &= \left\|\mathbf{X}_n(a)\theta(a) - \breve{\mathbf{Z}}_n(a)\theta(a) + \breve{\mathbf{Z}}_n(a)\theta(a) - \widehat{\mathbf{Z}}_n(a)\theta(a)\right\|_2^2 \\
&\leq 2\left\|\breve{\mathbf{Z}}_n(a)\theta(a) - \widehat{\mathbf{Z}}_n(a)\theta(a)\right\|_2^2 + 2\left\|\breve{\mathbf{Z}}_n(a)\theta(a) - \mathbf{X}_n(a)\theta(a)\right\|_2^2 \\
&= 2\left\|\breve{\mathbf{Z}}_n(a)\theta(a) - \widehat{\mathbf{Z}}_n(a)\theta(a)\right\|_2^2 + 2\left\|\breve{\mathcal{E}}_n(a)\theta(a)\right\|_2^2 \\
&= 2\left\|\mathbf{Z}_n(a)(\mathbf{P} - \widehat{\mathbf{P}}_n(a))\theta(a)\right\|_2^2 + 2\left\|\breve{\mathcal{E}}_n(a)\theta(a)\right\|_2^2 \\
&\leq 2L^2\sigma_1(\mathbf{Z}_n(a))^2\left\|\mathbf{P} - \widehat{\mathbf{P}}_n(a)\right\|_{op}^2 + 2L^2\left\|\breve{\mathcal{E}}_n\right\|_{op}^2,
\end{aligned}$$

where we have applied the parallelogram inequality to obtain the first inequality. □

**Lemma D.3.** *Suppose $(\xi_n)_{n\geq 1}$ is a sequence of independent, $\eta$-subGaussian random variables. Then, for any $\delta \in (0,1)$, we have simultaneously for all $n \geq 1$*

$$\left\|\Xi_n\right\|_2^2 \leq 6\eta^2\sqrt{2n\ell_\delta(n)} + 10\eta^2\ell_\delta(n) + n\alpha,$$

*where we recall $\Xi_n := (\xi_1, \ldots, \xi_n)^\top$.*

*Proof.* First, observe that by the appendix of Honorio and Jaakkola [49], since $\xi_n$ is $\eta$-subGaussian, $\xi_n^2 - \mathbb{E}\xi_n^2$ is $(4\sqrt{2}\eta^2, 4\eta^2)$-subExponential, and thus by our discussion in Appendix A $\xi_n^2 - \mathbb{E}\xi_n^2$ is also $(4\sqrt{2}\eta^2, 4\eta^2)$-subGamma. Further, by Assumption 3.3, we have $\sum_{m=1}^n \mathbb{E}\xi_m^2 \leq n\alpha$. Recall that Lemma A.2 yields that, with probability at least $1 - \delta$,

$$\sum_{m=1}^n \xi_m^2 - \mathbb{E}\xi_m^2 \leq 6\eta^2\sqrt{2n\ell_\delta(n)} + 10\eta^2\ell_\delta(n).$$

Piecing everything together, we have that

$$\|\Xi_n\|_2^2 = \sum_{m=1}^n \xi_m^2$$

$$= \sum_{m=1} \xi_m^2 - \mathbb{E}\xi_m^2 + \sum_{m=1}^n \mathbb{E}\xi_m^2$$

$$\leq 6\eta^2\sqrt{2n\ell_\delta(n)} + 10\eta^2\ell_\delta(n) + n\alpha.$$

$\square$

We now mention the finishing steps in proving Theorem 4.4, carrying over from where the proof in the paper ended. We see, with the previously addressed inequalities that, with probability at least $1 - O(A\delta)$, simultaneously for all $n \geq n_0$ and $a \in [A]$,

$$\left\|\widehat{\theta}_n(a) - \theta(a)\right\|_2^2 \leq \frac{6L^2U_n^2}{\sigma_r(\mathbf{Z}_n(a))^2} + \frac{2}{\sigma_r(\mathbf{Z}_n(a))^2}(T_1 + T_2)$$

$$= \frac{6L^2U_n^2}{\sigma_r(\mathbf{Z}_n(a))^2} + \frac{2}{\sigma_r(\mathbf{Z}_n(a))^2}\left[32\rho L^2 + 64\eta^2\left(\log\left(\frac{A}{\delta}\right) + r\log\left(1 + \frac{\sigma_1(\mathbf{Z}_n)^2}{\rho}\right)\right)\right.$$

$$\left. + 16L^2U_n^2 + 6\eta^2\sqrt{2c_n(a)\ell_\delta(c_n(a))} + 10\eta^2\ell_\delta(c_n(a)) + 6c_n(a)\alpha + + \frac{108L^2\sigma_1(\mathbf{Z}_n(a))^2U_n^2}{\sigma_r(\mathbf{Z}_n(a))^2} + 18L^2U_n^2\right]$$

$$= \frac{6L^2}{\widehat{\mathrm{snr}}_n(a)^2} + \frac{L^2}{\widehat{\mathrm{snr}}_n(a)^2}\left[68 + \frac{216\sigma_1(\mathbf{Z}_n(a))^2}{\sigma_r(\mathbf{Z}_n(a))^2}\right] + \frac{2}{\sigma_r(\mathbf{Z}_n(a))^2}\left[32\rho L^2\right.$$

$$\left. + 64\eta^2\left(\log\left(\frac{A}{\delta}\right) + r\log\left(1 + \frac{\sigma_1(\mathbf{Z}_n(a))^2}{\rho}\right)\right) + 6\eta^2\sqrt{2c_n(a)\ell_\delta(c_n(a))} + 10\eta^2\ell_\delta(c_n(a)) + 6c_n(a)\alpha\right]$$

# E    Proof of Theorem 4.1

In this section, we prove Theorem 4.1. All that is required in proving this bound is simplifying the results of Theorem 4.4, given we allow ourselves slack to control the bound up to universal constants and poly-logarithmic factors.

***Proof of Theorem 4.1.*** Recall that by Theorem 4.4, we have with probability at least $1 - O(A\delta)$, simultaneously for all $n \geq n_0$

$$\left\|\widehat{\theta}_n(a) - \theta(a)\right\|_2^2 \leq \frac{L^2}{\widehat{\mathrm{snr}}_n(a)^2}\left[74 + 216\kappa(\mathbf{Z}_n(a))^2\right] + \frac{2\mathrm{err}_n(a)}{\sigma_r(\mathbf{Z}_n(a))^2}.$$

where in the above we define the "error" term $\mathrm{err}_n(a)$ to be

$$\mathrm{err}_n(a) := \underbrace{24\rho L^2 + 64\eta^2\left(\log\left(\frac{A}{\delta}\right) + r\log\left(1 + \frac{\sigma_1(\mathbf{Z}_n(a))^2}{\rho}\right)\right)}_{T_1}$$

$$+ \underbrace{6\eta^2\sqrt{2c_n(a)\ell_\delta(c_n(a))} + 10\eta^2\ell_\delta(c_n(a)) + 6c_n(a)\alpha}_{T_2}.$$

First, by the second part of Lemma B.6 coupled with the assumption that $\mathrm{snr}_n(a) \geq 2$ for all $n \geq n_0$ and $a \in [A]$, we have that $\mathrm{snr}_n(a) = \Theta(\widehat{\mathrm{snr}}_n(a))$. Further, by this same result, we have that $\kappa(\mathbf{Z}_n(a)) = \Theta(\kappa(\mathbf{X}_n(a)))$. From this, it is clear that $\frac{L^2}{\widehat{\mathrm{snr}}_n(a)^2} \left[74 + 216\kappa(\mathbf{Z}_n(a))^2\right] = O\left(\frac{1}{\mathrm{snr}_n(a)^2}\kappa(\mathbf{X}_n(a))^2\right)$. What remains is to show that $\frac{2\mathrm{err}_n(a)}{\sigma_r(\widehat{\mathbf{Z}}_n(a))^2} = \widetilde{O}\left(\frac{1}{\mathrm{snr}_n(a)^2}\right)$. To this end, it suffices to show that $\mathrm{err}_n(a) = \widetilde{O}(n+d)$. But this is trivial, as it is clear that $T_1 = \widetilde{O}(1)$ and $T_2 = \widetilde{O}(c_n(a)) = \widetilde{O}(n+d)$. Thus, we have proved the desired result.

$\square$

# F  Equivalent formulations of ridge regression

We begin by discussing properties and equivalent formulations of ridge regression, as the estimate produced by (regularized) PCR, $\widehat{\theta}_n(a)$, is precisely the ridge estimate of the unknown parameter $\theta(a)$ when restricted the subspace associated with the projection matrix $\widehat{\mathbf{P}}_n$.

**Fact F.1** (**Ridge regression formulation**). *Let $\widehat{W}_n$ be the subspace associated with the projection matrix $\widehat{\mathbf{P}}_n$. Then, $\widehat{\theta}_n(a)$ satisfies*

$$\widehat{\theta}_n(a) = \arg\min_{\theta \in \widehat{W}_n} \left\{ \|\mathbf{Z}_n(a)\theta - \mathbf{Y}_n(a)\|_2^2 + \frac{\rho}{2}\|\theta\|_2^2 \right\}.$$

*That is, $\widehat{\theta}_n(a)$ is the solution to $\rho$-regularized ridge regression when estimates are restricted to $\widehat{W}_n$.*

Ridge regression may also be represented in the following, constrained optimization format.

**Fact F.2** (**Constrained formulation of ridge regression**). *Let $\widehat{W}_n$ be the subspace associated with the projection matrix $\widehat{\mathbf{P}}_n$. Then, $\widehat{\theta}_n(a)$ satisfies*

$$\widehat{\theta}_n(a) = \arg\min_{\theta \in \widehat{W}_n : \|\theta\|_2 \leq R_\rho} \|\mathbf{Z}_n(a)\theta - \mathbf{Y}_n(a)\|_2^2,$$

*where $R_\rho$ is some constant only depending on $\rho$.*

The larger $\rho$ is, the smaller $R_\rho$ must become. Since we know $\|\theta(a)\|_2 \leq L$, for all $a \in [A]$, throughout the main body and appendix of this paper, we assume that $\rho$ is chosen to be sufficiently small such that our estimates $\widehat{\theta}_n(a)$ satisfy $\|\widehat{\theta}_n(a)\|_2 \leq L$, i.e. we select $\rho > 0$ satisfying $R_\rho \leq L$.

# G  Proofs for application to panel data

**Lemma G.1** (**Reformulation of average expected post-intervention outcome**). *Under Assumption 5.1 and Assumption 5.2, the average expected post-intervention outcome of unit $n$ under intervention $a$ may be written as*

$$\mathbb{E}[\bar{Y}_{n,post}^{(a)}] = \frac{1}{T - T_0} \cdot \langle \theta(a), \mathbb{E}[Y_{n,pre}] \rangle,$$

*for some slope vector $\theta(a) \in \mathbb{R}^{T_0}$.*

*Proof.* Similar observations have been made in [45, 75]. For completeness, we include the proof here as well. From Assumption 5.1 and Definition 5.3,

$$\mathbb{E}[\bar{Y}_{n,post}^{(a)}] = \frac{1}{T - T_0} \cdot \left\langle \sum_{t=T_0+1}^{T} U_t^{(a)}, V_n \right\rangle.$$

Applying Assumption 5.2, we see that

$$\mathbb{E}[\bar{Y}_{n,post}^{(a)}] = \frac{1}{T - T_0} \cdot \left\langle \sum_{t=1}^{T_0} \theta(a)_t \cdot U_t^{(0)}, V_n \right\rangle$$

for some $\theta(a) = [\theta_1(a), \ldots, \theta_{T_0}(a)]^T \in \mathbb{R}^{T_0}$. $\square$

The following lemma follows straightforwardly from translating the notation of Theorem 4.4 to the panel data setting. Note that in the panel data setting, $d = T_0$, $\eta = \sigma\sqrt{T - T_0}$, and $\alpha = \sigma^2(T - T_0)$. Additionally, Assumption 3.1 is satisfied with $\gamma = \sigma^2$.

**Lemma G.2.** *Let $\delta \in (0, 1)$ be an arbitrary confidence parameter. Let $\rho > 0$ be chosen to be sufficiently small, as detailed in Appendix F. Further, assume that there is some $n_0 \geq 1$ such that $\mathrm{rank}(\mathbf{X}_{n_0}) = r$ and $\mathrm{snr}_n \geq 2$ for all $n \geq n_0$. Then, with probability at least $1 - O(A\delta)$, simultaneously for all actions $a \in [A]$ and time steps $n \geq n_0$, we have*

$$\left\| \widehat{\theta}_n(a) - \theta(a) \right\|_2^2 \leq \frac{L^2}{\widehat{\mathrm{snr}}_n(a)^2} \left[ 74 + 216\kappa(\mathbf{Z}_n(a))^2 \right] + \frac{2(T - T_0)\mathrm{err}_n(a)}{\sigma_r(\mathbf{Z}_n(a))^2},$$

*where $\kappa(\mathbf{Z}_n(a)) := \frac{\sigma_1(\mathbf{Z}_n(a))}{\sigma_r(\mathbf{Z}_n(a))}$, $\|\theta(a)\|_2 \leq L$, and in the above we define the "error" term $\mathrm{err}_n(a)$ to be*

$$\mathrm{err}_n(a) := \frac{32\rho L^2}{T - T_0} + 64\sigma^2 \left( \log\left(\frac{A}{\delta}\right) + r\log\left(1 + \frac{\sigma_1(\mathbf{Z}_n(a))^2}{\rho}\right) \right)$$
$$+ 6\sigma^2\sqrt{2c_n(a)\ell_\delta(c_n(a))} + 10\sigma^2\ell_\delta(c_n(a)) + 6\sigma^2 c_n(a).$$

**Theorem G.3 (Prediction error of average expected post-intervention outcome).** *Let $\delta \in (0, 1)$ be an arbitrary confidence parameter and $\rho > 0$ be chosen to be sufficiently small, as detailed in Appendix F. Further, assume that Assumptions 5.1 and 5.2 are satisfied, there is some $n_0 \geq 1$ such that $\mathrm{rank}(\mathbf{X}_{n_0}) = r$, and $\mathrm{snr}_n(a) \geq 2$ for all $n \geq n_0$. Then with probability at least $1 - O(A\delta)$, simultaneously for all interventions $a \in \{0, \dots, A - 1\}$*

$$|\widehat{\mathbb{E}}\bar{Y}_{n,post}^{(a)} - \mathbb{E}\bar{Y}_{n,post}^{(a)}| \leq \frac{3\sqrt{T_0}}{\widehat{\mathrm{snr}}_n(a)} \left( \frac{L(\sqrt{74} + 12\sqrt{6}\kappa(\mathbf{Z}_n(a)))}{(T - T_0) \cdot \widehat{\mathrm{snr}}_n(a)} + \frac{\sqrt{2\mathrm{err}_n(a)}}{\sqrt{T - T_0} \cdot \sigma_r(\mathbf{Z}_n(a))} \right)$$
$$+ \frac{2L\sqrt{24T_0}}{(T - T_0) \cdot \widehat{\mathrm{snr}}_n(a)} + \frac{12L\kappa(\mathbf{Z}_n(a))\sqrt{3T_0}}{(T - T_0) \cdot \widehat{\mathrm{snr}}_n(a)} + \frac{2\sqrt{\mathrm{err}_n(a)}}{\sqrt{T - T_0} \cdot \sigma_r(\mathbf{Z}_n(a))}$$
$$+ \frac{L\sigma\sqrt{\log(A/\delta)}}{\sqrt{T - T_0}} + \frac{L\sigma\sqrt{74\log(A/\delta)}}{\widehat{\mathrm{snr}}_n(a)\sqrt{T - T_0}} + \frac{12\sigma\kappa(\mathbf{Z}_n(a))\sqrt{6\log(A/\delta)}}{\widehat{\mathrm{snr}}_n(a)\sqrt{T - T_0}}$$
$$+ \frac{\sigma\sqrt{2\mathrm{err}_n(a)\log(A/\delta)}}{\sigma_r(\mathbf{Z}_n(a))},$$

*where $\widehat{\mathbb{E}}\bar{Y}_{n,post}^{(a)} := \frac{1}{T - T_0} \cdot \langle\widehat{\theta}_n(a), Y_{n,pre}\rangle$ is the estimated average post-intervention outcome for unit $n$ under intervention $a$.*

*Proof.*

$$\widehat{\mathbb{E}}\bar{Y}_{n,post}^{(a)} - \mathbb{E}\bar{Y}_{n,post}^{(a)} := \frac{1}{T - T_0}(\langle\widehat{\theta}_n(a), Y_{n,pre}\rangle - \langle\theta(a), Y_{n,pre}\rangle)$$
$$= \frac{1}{T - T_0}(\underbrace{\langle\widehat{\theta}_n(a) - \theta(a), \mathbb{E}Y_{n,pre}\rangle}_{T_1} + \underbrace{\langle\theta(a), \epsilon_{n,pre}\rangle}_{T_2}$$
$$+ \underbrace{\langle\widehat{\theta}_n(a) - \theta(a), \epsilon_{n,pre}\rangle}_{T_3})$$

We begin by bounding $T_1$. By assumption we have that $\mathbb{E}Y_{n,pre} \in \mathrm{span}(\mathbb{E}Y_{1,pre}, \dots, \mathbb{E}Y_{n-1,pre})$ for all $n \geq n_0$. Therefore,

$$\langle\widehat{\theta}_n(a) - \theta(a), \mathbb{E}Y_{n,pre}\rangle = \langle\widehat{\theta}_n(a) - \theta(a), \mathbb{E}Y_{n,pre}\mathbf{P}\rangle$$
$$= \langle\mathbf{P}\widehat{\theta}_n(a) - \theta(a), \mathbb{E}Y_{n,pre}\rangle$$
$$\leq \|\mathbb{E}Y_{n,pre}\|_2\|(\mathbf{P} - \widehat{\mathbf{P}}_n(a) + \widehat{\mathbf{P}}_n(a))(\widehat{\theta}_n(a) - \theta(a))\|_2$$
$$\leq \underbrace{\sqrt{T_0} \cdot \|(\mathbf{P} - \widehat{\mathbf{P}}_n(a))(\widehat{\theta}_n(a) - \theta(a))\|_2}_{T_{1.1}} + \underbrace{\sqrt{T_0} \cdot \|\widehat{\mathbf{P}}_n(a)(\widehat{\theta}_n(a) - \theta(a))\|_2}_{T_{1.2}}$$

By Lemma B.4 and the second part of Lemma B.6, term $T_{1.1}$ may be upper-bounded as

$$
\begin{aligned}
T_{1.1} &\leq \sqrt{T_0} \cdot \|\mathbf{P} - \widehat{\mathbf{P}}_n(a)\|_{op} \cdot \|\widehat{\theta}_n(a) - \theta(a)\|_2 \\
&\leq \sqrt{T_0} \cdot \frac{\sqrt{4\beta \left(3\sqrt{n\ell_{\delta/2\mathcal{N}}(n)} + 5\ell_{\delta/2\mathcal{N}}(n)\right) + 4n\gamma}}{\sigma_r(\mathbf{X}_n(a))} \cdot \|\widehat{\theta}_n(a) - \theta(a)\|_2 \\
&\leq \sqrt{T_0} \cdot \frac{\sqrt{4\beta \left(3\sqrt{n\ell_{\delta/2\mathcal{N}}(n)} + 5\ell_{\delta/2\mathcal{N}}(n)\right) + 4n\gamma}}{\frac{2}{3}\sigma_r(\mathbf{Z}_n(a))} \cdot \|\widehat{\theta}_n(a) - \theta(a)\|_2 \\
&= \frac{3\sqrt{T_0}}{\widehat{\mathrm{snr}}_n(a)} \cdot \|\widehat{\theta}_n(a) - \theta(a)\|_2
\end{aligned}
$$

Applying Lemma G.2, we see that

$$
\begin{aligned}
\frac{1}{T - T_0} T_{1.1} &\leq \frac{3\sqrt{T_0}}{(T - T_0) \cdot \widehat{\mathrm{snr}}_n(a)} \left(\frac{L^2}{\widehat{\mathrm{snr}}_n(a)^2} \left[74 + 216\kappa(\mathbf{Z}_n(a))^2\right] + \frac{2(T - T_0)\mathrm{err}_n(a)}{\sigma_r(\mathbf{Z}_n(a))^2}\right)^{1/2} \\
&\leq \frac{3\sqrt{T_0}}{(T - T_0) \cdot \widehat{\mathrm{snr}}_n(a)} \left(\frac{L}{\widehat{\mathrm{snr}}_n(a)} \sqrt{74 + 216\kappa(\mathbf{Z}_n(a))^2} + \frac{\sqrt{2(T - T_0)\mathrm{err}_n(a)}}{\sigma_r(\mathbf{Z}_n(a))}\right) \\
&= \frac{3\sqrt{T_0}}{\widehat{\mathrm{snr}}_n(a)} \left(\frac{L(\sqrt{74} + 12\sqrt{6}\kappa(\mathbf{Z}_n(a)))}{(T - T_0) \cdot \widehat{\mathrm{snr}}_n(a)} + \frac{\sqrt{2\mathrm{err}_n(a)}}{\sqrt{T - T_0} \cdot \sigma_r(\mathbf{Z}_n(a))}\right).
\end{aligned}
$$

Turning our attention to $T_{1.2}$ and using a line of reasoning nearly identical to equations (1), (2), (3) in the proof of Theorem 4.4, we get that with probability at least $1 - \mathcal{O}(A\delta)$,

$$
\begin{aligned}
\frac{1}{T - T_0} T_{1.2} &\leq \frac{2\sqrt{T_0}}{(T - T_0) \cdot \sigma_r(\mathbf{Z}_n(a))} \left(\left\|\widehat{\mathbf{Z}}_n(a)\widehat{\theta}_n(a) - \mathbf{X}_n(a)\theta(a)\right\|_2^2 + \left\|\mathbf{X}_n(a)\theta(a) - \widehat{\mathbf{Z}}_n(a)\theta(a)\right\|_2^2\right)^{1/2} \\
&\leq \frac{2\sqrt{T_0}}{(T - T_0) \cdot \sigma_r(\mathbf{Z}_n(a))} \left(32\rho L^2 + 64\sigma^2(T - T_0)\left(\log\left(\frac{A}{\delta}\right) + r\log\left(1 + \frac{\sigma_1(\mathbf{Z}_n(a))^2}{\rho}\right)\right)\right) \\
&\quad + 16L^2 U_n^2 + 6\sigma^2(T - T_0)\sqrt{2c_n(a)\ell_\delta(c_n(a))} + 10\sigma^2(T - T_0)\ell_\delta(c_n(a)) \\
&\quad + 6\sigma^2(T - T_0)c_n(a) + \frac{108L^2\sigma_1(\mathbf{Z}_n(a))^2 U_n^2}{\sigma_r(\mathbf{Z}_n(a))^2} + 18L^2 U_n^2\Bigg)^{1/2} \\
&= \frac{2\sqrt{T_0}}{(T - T_0) \cdot \sigma_r(\mathbf{Z}_n(a))} \left(34L^2 U_n^2 + \frac{108L^2\sigma_1(\mathbf{Z}_n(a))^2 U_n^2}{\sigma_r(\mathbf{Z}_n(a))^2} + (T - T_0)\mathrm{err}_n(a)\right)^{1/2} \\
&\leq \frac{2LU_n\sqrt{24T_0}}{(T - T_0) \cdot \sigma_r(\mathbf{Z}_n(a))} + \frac{12L\kappa(\mathbf{Z}_n(a))U_n\sqrt{3T_0}}{(T - T_0) \cdot \sigma_r(\mathbf{Z}_n(a))} + \frac{2\sqrt{\mathrm{err}_n(a)}}{\sqrt{T - T_0} \cdot \sigma_r(\mathbf{Z}_n(a))} \\
&= \frac{2L\sqrt{24T_0}}{(T - T_0) \cdot \widehat{\mathrm{snr}}_n(a)} + \frac{12L\kappa(\mathbf{Z}_n(a))\sqrt{3T_0}}{(T - T_0) \cdot \widehat{\mathrm{snr}}_n(a)} + \frac{2\sqrt{\mathrm{err}_n(a)}}{\sqrt{T - T_0} \cdot \sigma_r(\mathbf{Z}_n(a))}
\end{aligned}
$$

Putting our bounds for $T_{1.1}$ and $T_{1.2}$ together, we get that

$$
\begin{aligned}
\frac{\langle\widehat{\theta}_n(a) - \theta(a), \mathbb{E}Y_{n,pre}\rangle}{T - T_0} &\leq \frac{3\sqrt{T_0}}{\widehat{\mathrm{snr}}_n(a)} \left(\frac{L(\sqrt{74} + 12\sqrt{6}\kappa(\mathbf{Z}_n(a)))}{(T - T_0) \cdot \widehat{\mathrm{snr}}_n(a)} + \frac{\sqrt{2\mathrm{err}_n(a)}}{\sqrt{T - T_0} \cdot \sigma_r(\mathbf{Z}_n(a))}\right) \\
&\quad + \frac{2L\sqrt{24T_0}}{(T - T_0) \cdot \widehat{\mathrm{snr}}_n(a)} + \frac{12L\kappa(\mathbf{Z}_n(a))\sqrt{3T_0}}{(T - T_0) \cdot \widehat{\mathrm{snr}}_n(a)} + \frac{2\sqrt{\mathrm{err}_n(a)}}{\sqrt{T - T_0} \cdot \sigma_r(\mathbf{Z}_n(a))}
\end{aligned}
$$

Next we bound $T_2$. Note that $\langle\theta(a), \epsilon_{n,pre}\rangle$ is a $\|\theta(a)\|_2\sqrt{T - T_0}\sigma$-subGaussian random variable. Therefore via a Hoeffding bound, simultaneously for all actions $a \in [A]$, with probability at least $1 - \mathcal{O}(A\delta)$,

$$
\frac{\langle\theta(a), \epsilon_{n,pre}\rangle}{T - T_0} \leq L\sigma\sqrt{\frac{\log(A/\delta)}{T - T_0}}
$$

Similarly for $T_3$, $\langle\widehat{\theta}_n(a)-\theta(a),\epsilon_{n,pre}\rangle$ is a $\|\widehat{\theta}_n(a)-\theta(a)\|_2\sqrt{T-T_0}\sigma$-subGaussian random variable which, after applying a Hoeffding bound and our bound on $\|\widehat{\theta}_n(a)-\theta(a)\|_2$, becomes

$$
\begin{aligned}
\frac{\langle\widehat{\theta}_n(a)-\theta(a),\epsilon_{n,pre}\rangle}{T-T_0} &\leq \sigma\sqrt{\frac{\log(A/\delta)}{T-T_0}}\left(\frac{L^2}{\widehat{\mathrm{snr}}_n(a)^2}\left[74+216\kappa(\mathbf{Z}_n(a))^2\right]+\frac{2(T-T_0)\mathrm{err}_n(a)}{\sigma_r(\mathbf{Z}_n(a))^2}\right)^{1/2}\\
&\leq \sigma\sqrt{\frac{\log(A/\delta)}{T-T_0}}\left(\frac{\sqrt{74}L}{\widehat{\mathrm{snr}}_n(a)}+\frac{12\sqrt{6}\kappa(\mathbf{Z}_n(a))}{\widehat{\mathrm{snr}}_n(a)}+\frac{\sqrt{2(T-T_0)\mathrm{err}_n(a)}}{\sigma_r(\mathbf{Z}_n(a))}\right)\\
&= \frac{L\sigma\sqrt{74\log(A/\delta)}}{\widehat{\mathrm{snr}}_n(a)\sqrt{T-T_0}}+\frac{12\sigma\kappa(\mathbf{Z}_n(a))\sqrt{6\log(A/\delta)}}{\widehat{\mathrm{snr}}_n(a)\sqrt{T-T_0}}+\frac{\sigma\sqrt{2\mathrm{err}_n(a)\log(A/\delta)}}{\sigma_r(\mathbf{Z}_n(a))}
\end{aligned}
$$

Putting everything together, we see that with probability at least $1-O(A\delta)$,

$$
\begin{aligned}
|\widehat{\mathbb{E}}\bar{Y}_{n,post}^{(a)}-\mathbb{E}\bar{Y}_{n,post}^{(a)}| &\leq \frac{3\sqrt{T_0}}{\widehat{\mathrm{snr}}_n(a)}\left(\frac{L(\sqrt{74}+12\sqrt{6}\kappa(\mathbf{Z}_n(a)))}{(T-T_0)\cdot\widehat{\mathrm{snr}}_n(a)}+\frac{\sqrt{2\mathrm{err}_n(a)}}{\sqrt{T-T_0}\cdot\sigma_r(\mathbf{Z}_n(a))}\right)\\
&+\frac{2L\sqrt{24T_0}}{(T-T_0)\cdot\widehat{\mathrm{snr}}_n(a)}+\frac{12L\kappa(\mathbf{Z}_n(a))\sqrt{3T_0}}{(T-T_0)\cdot\widehat{\mathrm{snr}}_n(a)}+\frac{2\sqrt{\mathrm{err}_n(a)}}{\sqrt{T-T_0}\cdot\sigma_r(\mathbf{Z}_n(a))}\\
&+\frac{L\sigma\sqrt{\log(A/\delta)}}{\sqrt{T-T_0}}+\frac{L\sigma\sqrt{74\log(A/\delta)}}{\widehat{\mathrm{snr}}_n(a)\sqrt{T-T_0}}+\frac{12\sigma\kappa(\mathbf{Z}_n(a))\sqrt{6\log(A/\delta)}}{\widehat{\mathrm{snr}}_n(a)\sqrt{T-T_0}}\\
&+\frac{\sigma\sqrt{2\mathrm{err}_n(a)\log(A/\delta)}}{\sigma_r(\mathbf{Z}_n(a))}
\end{aligned}
$$

Applying Assumption 4.2, the expression simplifies to

$$
\begin{aligned}
|\widehat{\mathbb{E}}\bar{Y}_{n,post}^{(a)}-\mathbb{E}\bar{Y}_{n,post}^{(a)}| &= \widetilde{O}\left(\frac{r\sqrt{T_0}}{\sqrt{T_0\wedge n}}\left(\frac{Lr}{(T-T_0)\sqrt{T_0\wedge n}}+\frac{r}{\sqrt{(T-T_0)(T_0\wedge n)}}\right)\right.\\
&+\frac{Lr\sqrt{T_0}}{(T-T_0)\sqrt{T_0\wedge n}}+\frac{Lr\sqrt{T_0}}{(T-T_0)\sqrt{T_0\wedge n}}+\frac{r}{\sqrt{(T-T_0)(T_0\wedge n)}}\\
&\left.+\frac{L}{\sqrt{T-T_0}}+\frac{Lr}{\sqrt{(T-T_0)(T_0\wedge n)}}+\frac{r}{\sqrt{(T-T_0)(T_0\wedge n)}}+\frac{r}{\sqrt{T_0\wedge n}}\right).
\end{aligned}
$$

Under the assumptions that $T_0\leq\frac{1}{2}T$ and $r\leq\sqrt{T_0\wedge n}$, we get that

$$
|\widehat{\mathbb{E}}\bar{Y}_{n,post}^{(a)}-\mathbb{E}\bar{Y}_{n,post}^{(a)}| = \widetilde{O}\left(\frac{L}{\sqrt{T-T_0}}+\frac{r}{\sqrt{T_0\wedge n}}+\frac{r(L\vee 1)}{\sqrt{(T-T_0)(T_0\wedge n)}}\right)
$$

$\square$