# OpenReview forum: "Adaptive Principal Component Regression with Applications to Panel Data"
_NeurIPS.cc/2023/Conference — NeurIPS 2023 poster_

### Official Review · Reviewer_1mVx · 2023-06-30

**Soundness:** 4 excellent
**Presentation:** 2 fair
**Contribution:** 2 fair
**Rating:** 6
**Confidence:** 2

**Summary:**

Principal components regression (PCR) is common technique for handling data with error-in-variables. Prior studies had established bounds for PCR in traditional settings, but those bounds were not relevant for settings where data is collected adaptively. This paper provides computable bounds for the L2 error for coefficients in PCR specific to the adaptive data setting.

**Strengths:**

- The results of the paper are both new and technically sound.
- The bounds are computable under fairly reasonable assumptions.
- While I am not very familiar with martingale concentration, this paper provides further evidence that it is a reasonable approach for online learning problems.

**Weaknesses:**

- The paper satisfactorily describes how the new bounds compare to similar results for PCR in non-adaptive settings. However, there is limited discussion about why PCR might be a preferred methodology for online learning problems, beyond previous results in a generic error-in-variables setting.
- I commend the authors for including a section for an application of their work in an otherwise heavily theoretical paper. However, after some setup, the ultimate result appears to simply be a restatement of Theorem 4.1 / Corollary 4.3 but with some terms being a bit more concrete. The ultimate applicability of the result is not obvious.

**Questions:**

Presumably there are no other established results for models with error-in-variables and adaptive data. While the results are interesting, is there any point of comparison for other methodologies in this relatively narrowly defined area?

**Limitations:**

Limitations have been adequately addressed

---

> ### Author Rebuttal · Authors · 2023-08-09
>
> Thank you for taking the time to review our submission. Please find our replies to your questions/comments below.
>
> [*The paper satisfactorily describes how the new bounds compare to similar results for PCR in non-adaptive settings. However, there is limited discussion about why PCR might be a preferred methodology for online learning problems, beyond previous results in a generic error-in-variables setting.*]
>
> We would like to point out that our application to panel data is a salient application of online learning with noisy covariates. As we mention in the text, our framework may be thought of as a generalization of the synthetic interventions framework to adaptive settings, which itself is a generalization of the canonical synthetic control framework (a popular method both in theory and in practice).
>
> Moving away from panel data, another setting in which online learning from noisy data may arise is when the observed covariates need to be measured by a physical instrument. Since no measurement is perfectly accurate or exact, it may be helpful to explicitly take such measurement error into consideration when learning.
>
> In offline settings, almost all previous works in high-dimensional error-in-variables regression other than PCR requires oracle knowledge of the second moment of the noise matrix (e.g. [61, 70]). It has been shown recently that PCR both theoretically and empirically is able to deal with such error-in-variables without knowledge of the noise distribution, which is clearly valuable in practice.
>
> [*I commend the authors for including a section for an application of their work in an otherwise heavily theoretical paper. However, after some setup, the ultimate result appears to simply be a restatement of Theorem 4.1 / Corollary 4.3 but with some terms being a bit more concrete. The ultimate applicability of the result is not obvious.*]
>
> We would like to point out that we include a more thorough application of our adaptive bounds to panel data in Appendix F (see Theorem F.5 for a non-trivial application of our bounds). While we included this application in the appendix due to space constraints, we are happy to move more of these results to the main body in the revision.
>
>
> [*Presumably there are no other established results for models with error-in-variables and adaptive data. While the results are interesting, is there any point of comparison for other methodologies in this relatively narrowly defined area?*]
>
> To the best of our knowledge, there are no other results at the intersection of error-in-variables and adaptive data.
>
> Though it might seem as a narrowly-defined area, we formally establish that our results have important applications to the area of causal inference with panel data, in particular to the field of synthetic controls.  Specifically, our results allow us to estimate a variety of causal parameters of interest despite the intervention assignment being adaptive to the data itself, which clearly is true in many real-world settings. All previous formal results in the synthetic controls literature do not allow for adaptive data collection.
>
> Second, it has been shown that error-in-variables regression is a useful lens to think about differential privacy and regression ([5, 10]). In particular, if the covariates have been made differentially private by adding Gaussian noise, and we want to use these covariates for a downstream regression task, then this is an instance of  error-in-variables regression. And of course in many such settings, data is collected adaptively, and no prior results are able to tackle such a setting.
>
> We do indeed compare our results with the baseline of if the data is not collected adaptively (e.g. line 182), and we precisely quantify the penalty we pay in terms of sample complexity due to the adaptive data collection.

---

> > ### Comment · Reviewer_1mVx · 2023-08-16
> >
> > Thank you for your response. The additional context is appreciated, especially related to alternative methodologies, or lack thereof. Also, while I appreciate the results in Appendix F, I'm not sure that it alleviates my concern about the role of the application section in the overall paper.
> >
> > I have carefully evaluated your response, as well as the feedback provided by other reviewers. After due consideration, I have decided to retain my initial rating.

---

### Official Review · Reviewer_jhsg · 2023-07-06

**Soundness:** 4 excellent
**Presentation:** 3 good
**Contribution:** 3 good
**Rating:** 6
**Confidence:** 4

**Summary:**

This work studies the properties of principle component regression in the adaptive setting, where future observations may depend on the values of previously observed observations. The authors provide a time-uniform bounds on the L_2 error of online principal component regression that extend prior work to the adaptive setting. In the process they also remove a previously required assumption of soft sparsity, replacing it with a dependence on the signal to noise ratio. Finally, the authors describe how the results on PCR may be applied to the panel data setting which is commonly modeled via latent factor models.

**Strengths:**

* This is an interesting task setting. Adaptivity is common in practical settings, and this paper does a good job of providing a nice theoretical grounding for PCR in this setting.

* The authors do a nice job of providing clear, interpretable theoretical results. Decent motivation is also provided.

* The application to causal inference is a nice motivation (though it would have been nice to see an extended discussion)

**Weaknesses:**

Overall, while this is a nice set of results it would have been nice to see (a) a much more extended application of the theory to causal inference (or another relevant use case), and (b) the presence of _some_ amount of empirical evidence for the results. The authors also argue strongly that an advantage of the proposed approach is removing the soft sparsity assumption and replacing it with the SNR assumption. However, it's not clear to me in the adaptive case when we would expect for this to be advantageous. Shouldn't we expect some actions to be significantly favored during, e.g., a contextual bandit procedure that is pointed toward in the conclusion? It's also not entirely clear to me how useful the bounds provided in Theorem 4.4 are going to be outside of a theoretical setting. It would have been nice to have seen some empirical results to demonstrate their relative utility in lieu of the in-line proofs.

**Questions:**

It’s a little confusing why we would have separate noise assumptions over the treated and control series (assumptions 3.1 and 3.2, respectively). I recognize that 3.1 contains 3.2 but the assumption is considerably stronger. It’s not clear to me when we would be able to assume both the low rank assumption and bounded noise in practical settings.

The regularization justification seems a little strange. Aren’t we already imposing a regularization by restricting to the top $k$ principle components? It’s not entirely clear to me why imposing additional regularization is necessary in this setting.

Theorem 4.4 seems like it could be quite loose in empirical settings. Do the authors have empirical results that examine this?

Smaller items:

In Theorem 4.4 $L$ appears to be undefined in the main text.

It’s unclear within the main text whether the authors are employing assumption 3.1 or 3.2. Given that the latter is stronger, this should be stated either within the theorem text or directly before/after.

**Limitations:**

Yes.

---

> ### Author Rebuttal · Authors · 2023-08-09
>
> Thank you for taking the time to review our submission. Please find our responses to your questions/comments below.
>
>
> [*Overall, while this is a nice set of results it would have been nice to see (a) a much more extended application of the theory to causal inference (or another relevant use case)*]
>
> We would like to point out that we include a more thorough application of our adaptive bounds to panel data in Appendix F (see Theorem F.5 for a non-trivial application of our bounds). While we included this application in the appendix due to space constraints, we are happy to move more of these results to the main body in the revision.
>
>
> As for another use case, it has been shown that error-in-variables regression is a useful lens to think about differential privacy and regression ([5, 10]). In particular, if the covariates have been made differentially private by adding Gaussian noise, and we want to use these covariates for a downstream regression task, then this is an instance of  error-in-variables regression. And of course in many such settings, data is collected adaptively, and no prior results are able to tackle such a setting.
>
> [*Overall, while this is a nice set of results it would have been nice to see … (b) the presence of some amount of empirical evidence for the results.*]
>
> As mentioned in the global rebuttal, we chose to focus on theoretical depth for our adaptive bounds on PCR (Theorem 4.1, Theorem 4.4) and our application to panel data (Theorem F.5) in this submission, as a meaningful empirical comparison cannot be made with previous work since the constants in the non-adaptive bounds are not specified, and therefore the non-adaptive bound cannot be computed.
>
>
> [*The authors also argue strongly that an advantage of the proposed approach is removing the soft sparsity assumption and replacing it with the SNR assumption. However, it's not clear to me in the adaptive case when we would expect for this to be advantageous. Shouldn't we expect some actions to be significantly favored during, e.g., a contextual bandit procedure that is pointed toward in the conclusion?*]
>
> We would like to emphasize that the primary advantage of our bounds is that they hold in adaptive settings, whereas the bounds in previous work do not, and that we explicitly quantify the penalty paid for adaptivity (Line 182). With that being said, (1) it is reasonable to assume that in most contextual bandit tasks, different contexts have different optimal actions. Furthermore, (2) there is often sufficient diversity in the sequence of contexts encountered by a contextual bandit algorithm. For instance, in “A Contextual Bandit Bake-off”, Biette et al. find that when implementing popular contextual bandit algorithms on a wide variety of datasets, the greedy algorithm performed surprisingly well (2nd overall out of all algorithms implemented). The explanation given for the good performance of the greedy algorithm is that the inherent diversity in the distribution over contexts was oftentimes enough to provide sufficient exploration for good performance. Putting (1) and (2) together, we believe that in many practical contextual bandit settings, one or two actions will not be significantly favored over others.
>
>
> [*It’s a little confusing why we would have separate noise assumptions over the treated and control series (assumptions 3.1 and 3.2, respectively). I recognize that 3.1 contains 3.2 but the assumption is considerably stronger. It’s not clear to me when we would be able to assume both the low rank assumption and bounded noise in practical settings.*]
>
> We emphasize that Assumption 3.2 is not required for any of our results to hold, and that all theorems hold under Assumption 3.1. We will make this clearer in the next revision. Assumption 3.2 just allows us to make stronger statements (e.g. tighter constants with respect to the results obtained under Assumption 3.1) in some settings. We also note that the low rank and bounded noise assumptions are orthogonal to one another, as assuming one does not make it easier or harder for the other to hold.
>
>
> Furthermore, the stated bounds do not require specifying which assumption (either 3.1 or 3.2) holds, as this only makes a difference in the definition of the signal-to-noise ratio. If Assumption 3.1 (resp. Assumption 3.2) holds, one would use the definition of signal-to-noise with $U_n$ corresponding to the line below 193 (resp. line above 194).
>
>
> [*The regularization justification seems a little strange. Aren’t we already imposing a regularization by restricting to the top $k$ principle components? It’s not entirely clear to me why imposing additional regularization is necessary in this setting.*]
>
> As we note in Section 3, we choose to regularize since it is known that regularization increases the stability of regression-style algorithms in online learning settings. Since regularizing is a popular choice in online settings, by regularizing we are able to utilize existing online regression bounds from previous work. Such regularization is required for, e.g. the Method of Mixtures result of Abbasi-Yadkori et al. [3] (Lemma A.1).
>
>
> [*Theorem 4.4 seems like it could be quite loose in empirical settings. Do the authors have empirical results that examine this?*]
>
> Please see our above comments regarding the lack of an empirical evaluation.
>
>
> [*In Theorem 4.4 $L$ appears to be undefined in the main text.*]
>
> Thank you for pointing this out. $L$ is an upper-bound on the l2 norm of $\theta(a)$.
>
> [*It’s unclear within the main text whether the authors are employing assumption 3.1 or 3.2. Given that the latter is stronger, this should be stated either within the theorem text or directly before/after.*]
>
> Please see our above answer regarding your question on separate noise assumptions.

---

> > ### Comment · Reviewer_jhsg · 2023-08-21
> >
> > I would like to thank the authors for taking the time to respond in such a thoughtful way. My concerns have largely been addressed, and I am raising my score to reflect this, with my remaining concerns still largely centered on framing/applicability.

---

### Official Review · Reviewer_Dfi8 · 2023-07-08

**Soundness:** 2 fair
**Presentation:** 3 good
**Contribution:** 3 good
**Rating:** 4
**Confidence:** 4

**Summary:**

This paper studies theoretical properties of the principal component regression (PCR) when data are collected adaptively and are corrupted with measurement errors. In particular, it gives an upper bound of the estimation error of the regression coefficients of the PCR. The derived bounds are then used to study the estimation error of causal effects in panel data setting.

**Strengths:**

The paper is well written and the problem under study if of interest. The theoretical findings appear to be interesting.

**Weaknesses:**

1. There are some issues with the theoretical findings which I elaborate later.
2. There is no numerical evaluations to support the theoretical findings.
3. There is no real data application so that one can evaluate the practical usefulness of the derived bounds.

**Questions:**

1. Regarding the theoretical results.
 (a) The model is assumed to be $Y_n=<\theta(a_n),X_n>+\xi_n$ where $X_n\in W^*$ with $dim(W=r<d)$. The parameter vector $\theta(a)$ is a d-dimensional vector that also resides in $W^*$. This setting is sort of strange because I am not even sure $\theta(a)$ is identifiable in such a setting. Please clarify.
 (b) Related to the above question, in definition 3.4, the matrix $\hat\nu_n(a)$ should be of at most rank $k<d$ by definition. Is it even invertible?
(c) In Corollary 4.3, the convergence rate becomes faster when $d$ increases. This is a little counterintuitive. When d-increases, there are more parameters to be estimated and one would expect a slower convergence rate. Can you explain the intuition behind this finding?
(d) Corollary 4.3, in a special case, when $d=r$, no dimension reduction is needed and the model under study becomes a regular linear regression model. Then the convergence rate is $O(1)$. This is true in linear regression models with measurement errors since the bias in regression coefficient estimates does not vanish, see, e.g., Carroll et al. (1995). Does this mean the derived bound only applies for the case $d\to\infty$?

2. Regarding adaptivity. The authors emphasize that the data are collected adaptively, and hence can be correlated. However, the correlation issue is not adequately addressed in the theoretical proof.

(a) On page 4, lines 159-160, the authors introduce the notation $\mathbf P$ as the projection of $\mathbf Z_n$ onto the true subspace $W_*$. However, since rows of $\mathbf Z_n$ are adaptively collected with a stochastic action sequence $A_n=\{a_1,\cdots, a_n\}$. Therefore,  $\mathbf Z_n$ is dependent on $A_n$ and so should the $\mathbf P_{n,k}$. Therefore, $\mathbf P_{n,}$ should be written as $\mathbf P_{n,k} (A_n)$. And the authors claim that $P_{n,k} (A_n)\to P$. If $A_n$ is stochastic, what is the sample version $\hat{\mathbf P}_{n,k}$ converges to as $n\to\infty$? What is the exact definition of $P$?

(b) On page 5, line 184, the authors comment that "bounding $||\varepsilon_n(a)||_{op}$ is a nontrivial task as the rows of $\varepsilon_n(a)$ may be strongly correlated". However, the matrix $\varepsilon_n$ is a just stacked version of rows in $\varepsilon_n(a)$'s. In a special case where the first half of the rows all take action $a_1$ and the second half of rows all take action $a_2$. If rows in $\varepsilon_n(a_1)$ and $\varepsilon_n(a_2)$ are highly correlated, I don't see why rows in $\varepsilon_n$ are not highly correlated. I think the paper implicitly assumes that the measurement errors are independent of the actions taken. However, if this is the case, rows in $\varepsilon_n(a)$ would also be independent as well and the adaptivity wouldn't matter. Please clarify.

(c) Similar to the above question, in the proofs of Lemma A.2-A.4 and B.3, the measurement errors are assumed to be independent of each other. However, the paper emphasizes that the data are collected adaptively. This pretty much means that the measurement errors $\epsilon_i$'s are independent of actions taken. Again, the adaptivity wouldn't matter in this case.

3. There is no numerical evaluations to support the theoretical findings.

4. There is no real data application so that one can evaluate the practical usefulness of the derived bounds.

Reference:
1. Carroll, R. J., Ruppert, D., & Stefanski, L. A. (1995). Measurement error in nonlinear models (Vol. 105). CRC press.

---

> ### Author Rebuttal · Authors · 2023-08-09
>
> Thanks for reviewing our work; our replies are below.
>
>
> [*Reply to 3 & 4.*]
>
> See global rebuttal.
>
>
> [*Reply to 1(a)*]
>
> The assumption that $\theta(1), ..., \theta(A)$ lie in a low-dimensional subspace follows from the latent factor model that is ubiquitous in panel data. This can be seen in Lemma F.4; we are happy to provide more details if necessary. In the high-dimensional setting, the $\theta(a)$ need not be identifiable. As shown in existing work, the minimum l2-norm $\theta(a)$ consistent with observed data is, in fact, identifiable. This $\theta(a)$ is the unknown slope vector lying in the low dimensional subspace. The conclusions of this work establish that not only are the $\theta(a)$ identifiable, but also that they are consistently estimable in the large sample limit under reasonable assumptions on signal to noise ratio.
>
> [*Reply to 1(b)*]
>
> The matrix is not in general invertible, but this is not a problem as the restriction of $\widehat{V}_n(a)$ (viewed as a linear transformation) to the top r eigenspace (i.e. the subspace spanned by the top r eigenvectors) will be invertible given that the empirical signal-to-noise ratio is not zero. The vector being hit on the left by $\widehat{V}_n(a)$ in the construction of the estimate of $\theta(a)$ lies in the top r eigenspace by definition, and thus the action of $\widehat{V}_n(a)$ is uniquely defined. The use of $\widehat{V}_n(a)^{-1}$ could freely be replaced by the Moore-Penrose pseudo-inverse of $\widehat{V}_n(a)^+$ (which is defined on all of $R^d$, not just the top r eigenspace) with no effect on computation.
>
> [*Reply to 1(c)*]
>
> Interestingly, since the true parameter is rank-r (which we assume to be fixed) the growing dimension d is actually beneficial, as each column of the matrix can be thought of as a “repeated measurement” of the r latent factors (i.e. a linear combination of the r latent factors). This is in line with the matrix completion literature where how well the matrix is de-noised grows as $r / (\min{n, d})$, and thus requires both dimensions to be growing and that $r << \min{n, d}$. Indeed the PCA step of the PCR algorithm can be thought of as “de-noising” the noisily observed covariates as is done in matrix completion.
>
> [*Reply to 1(d)*]
>
> Our bounds in Theorems 4.1 and 4.4 are fully non-asymptotic (i.e. they provide finite-sample guarantees). In the case $d = r$, there is no low rank structure for PCR to exploit, and thus the algorithm is not useful in this setting. We emphasize the result presented in Corollary 4.3 holds in any scaling regime of the parameters $d, n,$ and $r$. For instance, the bound would hold if $d, r$ were fixed and $n$ were taken to grow towards infinity (setting of classical asymptotic statistics) or if $n, r$ were taken to grow with $d$ (the setting for modern high-dimensional statistics).
>
>
> [*Reply to 2(a)*]
>
>
> Under the assumptions of Theorem 4.1 and 4.4 (that $rank(X_n) = r$ for $n$ sufficiently large), the recovered projection matrix for the non-noisy data is unique, representing the unique orthogonal projection operator onto the true subspace $W^*$. This assumption can be viewed as requiring sufficiently diverse treatment assignments to units in the panel data setting.
>
>
> [*Reply to 2(b)*]
>
> If the case of bounding $\|\epsilon_n(a)\|$, the choice of action in round $n$ and the observed noise $\epsilon_n$ are correlated random variables (e.g. our choice of action or chosen treatment of an individual will be influenced by the noise present in covariates). More formally, suppose $n_1, \dots, n_k$ are the (random) rounds on which treatment $a$ was assigned. Conditioned on knowing that treatment $a$ was given on these rounds, the distributions of $\epsilon_1, \dots, \epsilon_n$ are no longer independent, and $E(\epsilon_n | A_n = a)$ may not be even equal to zero. Just because the $\epsilon_1, \dots, \epsilon_n$ are marginally independent does not imply they are conditionally independent.  For instance, we may have a treatment rule “assign treatment 1 if the first covariate exceeds 5, else assign treatment 0”. Thus we cannot apply concentration results that require that the rows of the matrix have mean 0 or that the rows are independent, as this will generally not be the case. The rows of $\epsilon_n$ are independent by assumption on the noise in covariates (there is no dependence on the action in the construction of $\epsilon_n$). In particular the rows are just i.i.d. random vectors under our assumptions.
>
>
> [*Reply to 2(c)*]
>
> The measurement errors could equivalently be taken to be subGaussian and mean 0 *conditioned* on the information observed up to time $n - 1$ (i.e. our results would still follow if we assume $E(\epsilon_n | F_{n  -1}) = 0$ and $\log E(\exp\{\langle \lambda, \epsilon_n\rangle\} | F_{n - 1}) \leq \frac{\|\lambda\|^2}{2}$, where $F_n$ is the $\sigma$-algebra generated by the observations made up to time $n$ (think of this as the “history” of the interaction up to time $n$). Martingale concentration naturally applies to this more general dependence structure (in which the noise in covariates is not necessarily independent). However to greatly simplify notation used in proofs, we assume independence. Note that while the noise in covariates in round $n$ is independent of actions taken in rounds $1, \dots, n - 1$, it is NOT independent of the action taken in round $n$ (the conditional distribution of $\epsilon_n$ given the observation $A_n$ may not even by mean zero). Moreover, the choice of covariates is not assumed to be independent. One simple way of viewing this is that nature uses the information accumulated up to (not including) time $n$ to generate the covariate $X_n$, and then adds as corruption an independent sample from a standard, multivariate normal distribution. The learner then uses this corrupted covariate to determine a course of treatment. The noise here is independent of previous observations and treatment, but not of the $n$th choice of treatment.

---

> > ### Comment · Reviewer_Dfi8 · 2023-08-21
> >
> > I would like to thank authors for detailed responses to my questions. But I am not convinced by the arguments, hence will keep my score as it is.

---

> > > ### Author Response · Authors · 2023-08-21
> > >
> > > We are happy to further clarify any questions the reviewer has, as it is not clear which specific concerns the reviewer still has. We believe the majority of our rebuttal was clarifying linear algebraic facts (such as properties of projection matrices), common assumptions made in the matrix completion/synthetic controls literature, and the role of adaptivity in our arguments. We are thus unsure what the reviewer finds unconvincing.

---

> > > > ### Comment · Reviewer_Dfi8 · 2023-08-21
> > > >
> > > > For question 1(c), fixing the rank r still looks strange. Instead of adding columns that are linear combinations of r latent factors, it is more likely in practice to add irrelevant noisy  columns, in which case the rank of X will increases as d increases. Fixing r seems unnatural to me.
> > > >
> > > > For questions 2(c), you still need to assume the assumption that the measurement errors are independent of the actions taken, correct? Since in the paper  you did not use the martingale conditions.

---

> > > > > ### Author Response · Authors · 2023-08-21
> > > > >
> > > > > *1. For question 1(c), fixing the rank r still looks strange. Instead of adding columns that are linear combinations of r latent factors, it is more likely in practice to add irrelevant noisy columns, in which case the rank of X will increases as d increases. Fixing r seems unnatural to me.*
> > > > >
> > > > > Even as more covariates are observed over time (that is, as new rows are added), the $r$-dimensional subspace spanned by the $d$-dimensional covariates remains fixed over time. In other words, the rowspan of $X_n$ (the matrix of covariates *without* measurement errors) is a fixed, $r$-dimensional subspace of $R^d$ for all time steps $n \geq n_0$. This sort of setting follows directly from the latent factor model. We emphasize that this latent factor model has been widely accepted as the go-to model in many econometric papers. In particular, those on panel data, principal component regression, and synthetic interventions/controls [1, 2, 3, 4, 7, 8]. Given that the importance these works have had in addressing practical, real-world econometric problems, we believe that the model studied is, in fact, of great practical relevance and reasonable to assume. If the reviewer is uncomfortable with assuming $r$ is known, we note there exist practically relevant heuristics for estimating $r$ (see [7], for instance).
> > > > >
> > > > > *2. For questions 2(c), you still need to assume the assumption that the measurement errors are independent of the actions taken, correct? Since in the paper you did not use the martingale conditions.*
> > > > >
> > > > > We emphasize that the errors/noise in covariates (that is the rows of the noisy covariate matrix $Z_n$) do NOT need to be independent of action taken, but that the error in the response (i.e. reward) is assumed to be independent of the action taken. Moreover, we believe the reviewer is mistaken, as in the paper we heavily leverage martingale analysis heavily to prove our results (see Appendix A for a detailed description of the results we use). In particular, we leverage the self-normalized concentration results of [5] and [6] throughout our work. These results directly apply to the more general (but more notationally cumbersome) noise structure defined in our first response (that of being conditionally sub-Gaussian conditioned on the natural filtration associated with observations up to time $n$).
> > > > >
> > > > >
> > > > > [1] Manuel Arellano and Bo Honore. Panel data models: Some recent developments. Handbook
> > > > > of Econometrics, 02 2000.
> > > > >
> > > > > [2] Anish Agarwal, Devavrat Shah, Dennis Shen, and Dogyoon Song. On robustness of principal
> > > > > component regression. Journal of the American Statistical Association, 116(536):1731–1745,
> > > > > 2021. doi: 10.1080/01621459.2021.1928513
> > > > >
> > > > > [3] Kung-Yee Liang and Scott L. Zeger. Longitudinal data analysis using generalized linear
> > > > > models. Biometrika, 73(1):13–22, 04 1986. ISSN 0006-3444. doi: 10.1093/biomet/73.1.13.
> > > > > URL https://doi.org/10.1093/biomet/73.1.13.
> > > > >
> > > > > [4] Manuel Arellano and Bo Honore. Panel data models: Some recent developments. Handbook
> > > > > of Econometrics, 02 2000.
> > > > >
> > > > > [5] Yasin Abbasi-Yadkori, Dávid Pál, and Csaba Szepesvári. Improved algorithms for linear
> > > > > stochastic bandits. Advances in neural information processing systems, 24, 2011.
> > > > >
> > > > > [6] Steven R Howard, Aaditya Ramdas, Jon McAuliffe, and Jasjeet Sekhon. Time-uniform,
> > > > > nonparametric, nonasymptotic confidence sequences. 2021.
> > > > >
> > > > > [7] Anish Agarwal, Devavrat Shah, and Dennis Shen. Synthetic interventions. arXiv preprint
> > > > > arXiv:2006.07691, 2020.
> > > > >
> > > > > [8] Alberto Abadie, Alexis Diamond, and Jens Hainmueller. Synthetic control methods for
> > > > > comparative case studies: Estimating the effect of california’s tobacco control program.
> > > > > Journal of the American statistical Association, 105(490):493–505, 2010.

---

> > > > > > ### Comment · Reviewer_Dfi8 · 2023-08-22
> > > > > >
> > > > > > For 1(c), I was talking about adding more columns, not rows.
> > > > > >
> > > > > > For 2(c), I meant that "the errors in the response" is assumed to be independent of the actions, which is a strong assumption. With such an assumption. The adaptivity achieved under such an assumption is rather limited.

---

> > > > > > > ### Author Response · Authors · 2023-08-22
> > > > > > >
> > > > > > > Thank you for your quick response.
> > > > > > >
> > > > > > > To clarify, for any given problem instance, the only parameter that grows is the number of data points (i.e., rows) that arrive, and the number of columns is fixed. This is why we based our previous response on “growing rows.” However, one can compare different families of problem instances with different dimensions. Our initial answer to question 1(c) already describes what happens if $d$ were to increase while $r$ remains fixed. Also, we believe that the reviewer’s comment about how rank can increase with added “noisy columns” cannot occur in our setting, since there is no noise in the (unobserved) latent covariate matrix $X$, which has a fixed rank. Finally, we remark that Corollary 4.3 is just one particular setting under which our bounds take a “nice” form. For a more general result which makes relatively few assumptions on the relationship between $r$, $n$, and $d$, see Theorem 4.1 or Theorem 4.4.
> > > > > > >
> > > > > > > Regarding 2(c), the difficulty in our setting comes from handling correlations between the noise in the covariates and the actions, which was not handed in previous work on principal component regression or synthetic control/intervention. Moreover, we do not believe that the errors in the response being independent of the actions is a strong assumption. Indeed, it is usually assumed in, e.g., linear bandit settings that the noise is conditionally subgaussian, and our results may be readily extended to handle conditionally subgaussian noise. (See our original answer to 2(c) for more details.) Finally, previous work on PCR and synthetic control/interventions also assumes that actions are independent from response error.

---

> > > > > > > > ### Comment · Reviewer_Dfi8 · 2023-08-22
> > > > > > > >
> > > > > > > > For 1c, adding "noisy columns" means adding a column that is not in the linear column space of the columns of X, which is a very realistic setting when one has a high dimensional regression problem. The estimation error typically increases as $d$ increases. Assuming a fix rank for X while d increases is not very realistic in practice. What I am saying is that the setting under consideration is not a very realistic one.
> > > > > > > >
> > > > > > > > For 2c, I quoted from the paper On page 5, line 184, the authors comment that "bounding $||\varepsilon_n(a)||_{op}$ is a nontrivial task as the rows of $\varepsilon_n(a)$ may be strongly correlated". Here, $\varepsilon_n(a)$ is the error in response.  And now your are claiming that   "the difficulty in our setting comes from handling correlations between the noise in the covariates and the actions". I am a bit confused here.

---

> > > > > > > > > ### Author Response · Authors · 2023-08-22
> > > > > > > > >
> > > > > > > > > *For 1c, adding "noisy columns" means adding a column that is not in the linear column space of the columns of X, which is a very realistic setting when one has a high dimensional regression problem. The estimation error typically increases as d increases. Assuming a fix rank for X while d increases is not very realistic in practice. What I am saying is that the setting under consideration is not a very realistic one.*
> > > > > > > > >
> > > > > > > > > We believe we have addressed the reviewer's concerns in 1(c) to the best of our abilities, but recommend that the reviewer consults our response above pertaining to online learning (a common, realistic setting in which $d$ is fixed and $n$ grows) and our responses to 1(c) and 1(d) in the initial rebuttal (which address high-dimensional setting in which $n, d,$ and $r$ all grow in unison).
> > > > > > > > >
> > > > > > > > > *For 2c, I quoted from the paper On page 5, line 184, the authors comment that "bounding $\|\epsilon_n(a)\|$is a nontrivial task as the rows of $\epsilon_n(a)$ may be strongly correlated". Here, $\epsilon_n(a)$ is the error in response. And now your are claiming that "the difficulty in our setting comes from handling correlations between the noise in the covariates and the actions". I am a bit confused here.*
> > > > > > > > >
> > > > > > > > > We believe the reviewer has misunderstood notation. Any quantity involving $\epsilon$ (e.g. the matrices $\mathcal{E}_n$ and $\mathcal{E}_n(a)$) pertains to *noise in covariates*, while quantities involving $\xi$ or $\Xi$ pertain to noise in responses (i.e. noise in rewards). We recommend the reviewer consult the problem setup section (Section 3.1) for further clarification. We hope this helps with the confusion.

---

> > > > > > > > > > ### Comment · Reviewer_Dfi8 · 2023-08-22
> > > > > > > > > >
> > > > > > > > > > For 1c, I still concerns about the practical feasibility of the setting.
> > > > > > > > > >
> > > > > > > > > > For 2c, then my original question was correct and let me reiterate my original question "(c) Similar to the above question, in the proofs of Lemma A.2-A.4 and B.3, the measurement errors are assumed to be independent of each other. However, the paper emphasizes that the data are collected adaptively. This pretty much means that the measurement errors  are independent of actions taken. "  I am not convinced by your argument about "measurement errors in covariates are not independent of actions taken".

---

> > > > > > > > > > > ### Author Response · Authors · 2023-08-22
> > > > > > > > > > >
> > > > > > > > > > > *[For 1c, I still concerns about the practical feasibility of the setting.]*
> > > > > > > > > > >
> > > > > > > > > > > We inherit the same low rank structure that is a common assumption in synthetic control methods and PCR; both of which are popular methods which are used in practice.
> > > > > > > > > > >
> > > > > > > > > > > *[For 2c, then my original question was correct and let me reiterate my original question "(c) Similar to the above question, in the proofs of Lemma A.2-A.4 and B.3, the measurement errors are assumed to be independent of each other. However, the paper emphasizes that the data are collected adaptively. This pretty much means that the measurement errors are independent of actions taken. " I am not convinced by your argument about "measurement errors in covariates are not independent of actions taken".]*
> > > > > > > > > > >
> > > > > > > > > > > To reiterate, we emphasize that the measurement errors in covariates do NOT need to be independent of the actions taken, which is not in contradiction of our applications of Lemmas A.2-A.4 and Lemma B.3. While these results do require independence, we only apply them to bound quantities which are indeed independent, not to bound action-dependent measurement errors. We are happy to answer any specific questions regarding these lemmas that the reviewer may have.

---

> > > > > > > > > > > > ### Comment · Reviewer_Dfi8 · 2023-08-22
> > > > > > > > > > > >
> > > > > > > > > > > > Isn't Lemma B.3 used to bound the operator norm of the measurement errors, which are assumed to be independent of (actions)?

---

> > > > > > > > > > > > > ### Author Response · Authors · 2023-08-22
> > > > > > > > > > > > >
> > > > > > > > > > > > > You are correct that we use Lemma B.3 to bound $||\mathcal{E}_n||^2$. The rows of $\mathcal{E}_n$ (the noise in the *entire* covariate matrix) are independent by construction; the rows of $\mathcal{E}_n(a)$ (the noise in covariates *in rounds where action $a$ was selected*) are not. This is because we allow the action $a_n$ to depend on the (noisy) observations up to round $n$. Also note that $||\mathcal{E}_n(a)||^2 \leq ||\mathcal{E}_n||^2$.

---

### Official Review · Reviewer_M8r1 · 2023-07-10

**Soundness:** 4 excellent
**Presentation:** 3 good
**Contribution:** 3 good
**Rating:** 6
**Confidence:** 3

**Summary:**

This study provides finite sample guarantees  for PCR regression in the online setting where covariates and interventions are chosen adaptively. The key technical contributions hinge on applying recent results on self-normalized martingale concentration to this problem. The authors further apply these novel bounds in the context of learning from panel data when interventions are assigned adaptively.

**Strengths:**

The paper is overall well written, with a good mix of intuition and technical detail. The theoretical findings around the self normalized martingale concentration application are insightful. The application to adaptive panel data setting illustrates the usefulness of this method well. Furthermore, this method enables some future work around noisy LinUCB bounds which I look forward to. In general, I found this paper to be engaging and believe that with a few additional adjustments, it has the potential to make a valuable contribution to the conference.

**Weaknesses:**

The problem addressed in this paper appears to be quite specific, and it is unclear how frequently online learning with noisy covariates occurs in practical scenarios. It would be beneficial to include concrete examples that illustrate situations where this problem could arise. Additionally, one notable concern is the absence of an experimental section in this work. While the paper primarily focuses on theoretical aspects, it would be valuable to have an experimental evaluation of the proposed bounds to further validate their effectiveness.

**Questions:**

* How common is the problem of learning with noisy covariates, especially in an online setting?
* While I don't see an explicit assumption, do the error terms have to be homoscedastic for the analysis to work?
* How strong are the assumptions 3.1-3.3 and how do they compare with those in similar literature?
* Could you provide some experimental evaluation?



**Limitations:**

The authors don't explicitly addressed the limitations of their work.

---

> ### Author Rebuttal · Authors · 2023-08-09
>
> Thank you for taking the time to review our submission. Please find our replies to your questions/comments below.
>
> [*How common is the problem of learning with noisy covariates, especially in an online setting?*]
>
> We would like to point out that our application to panel data is a salient application of online learning with noisy covariates. As we mention in the text, our framework may be thought of as a generalization of the synthetic interventions framework to adaptive settings, which itself is a generalization of the canonical synthetic control framework (and is a popular method for counterfactual estimation both in theory and in practice).
>
> Moving away from panel data, another setting in which online learning from noisy data may arise is when the observed covariates need to be measured by a physical instrument. Since no measurement is perfectly accurate or exact, it may be helpful to explicitly take such measurement error into consideration when learning.
>
> Finally, it has been shown that error-in-variables regression is a useful lens to think about differential privacy and regression ([5, 10]). In particular, if the covariates have been made differentially private by adding Gaussian noise, and we want to use these covariates for a downstream regression task, then this is an instance of  error-in-variables regression. And of course in many such settings, data is collected adaptively, and no prior results are able to tackle such a setting.
>
> [*While I don't see an explicit assumption, do the error terms have to be homoscedastic for the analysis to work?*]
>
> It suffices to consider error terms with the same upper bound on their variance, but can be heteroscedastic otherwise. The assumptions of either bounded or subGaussian noise both provide for a uniform upper bound on the variance of noise variables.
>
> [*How strong are the assumptions 3.1-3.3 and how do they compare with those in similar literature?*]
>
> Assumptions 3.1 and 3.3 are fairly mild and are standard in the literature on “error-in-variables” regression (see e.g. [7,8]). Assumption 3.2 is a stronger assumption (bounded random variables vs subGaussian). We emphasize that Assumption 3.2 is not required for any of our results to hold; it just allows us to make stronger statements (e.g. tighter constants) in some settings. We believe the setting of bounded noise is of high utility for applications of our bounds, as noise is often bounded in real-world settings (for instance, a given piece of information associated with an individual may be a percentage, which will always lie between 0 and 1).
>
>
> [*Could you provide some experimental evaluation?*]
>
>
> As stated in the global rebuttal, we chose to focus on theoretical depth for our adaptive bounds on PCR (Theorem 4.1, Theorem 4.4) and our application to panel data (Theorem F.5) in this submission, as a meaningful empirical comparison cannot be made with previous work since the constants in the non-adaptive bounds are not specified, and therefore the non-adaptive bound cannot be computed.

---

### Author Rebuttal · Authors · 2023-08-09

We would like to thank the reviewers for their helpful reviews. We have responded to each of your comments, and we hope to address any follow-up questions you may have in the discussion.

To summarize our main contributions, we provide the first time-uniform finite sample guarantees for principal component regression whenever data is collected adaptively. Additionally, we apply our results to the problem of estimating unit-specific treatment effects in panel data settings, where our methodology may be thought of as a generalization of the popular synthetic control framework to settings in which data is collected adaptively.

A common complaint from reviewers was that our submission contains no empirical results. We chose to focus on theoretical depth for our adaptive bounds on PCR (Theorem 4.1, Theorem 4.4) and our application to panel data (Theorem F.5) in this submission, as a meaningful empirical comparison cannot be made with previous work since the constants in the non-adaptive bounds are not specified, and therefore the non-adaptive bound cannot be computed.

---

### Decision · Program_Chairs · 2023-09-21

**Decision:**

Accept (poster)

**Comment:**

The reviewers are largely in agreement that the authors provide a technically strong theoretical contribution to the literature on principal component regression and and to the synthetic control literature in causal inference. There is however an overall feeling among reviewers that the setting analyzed is quite specific and there are some assumptions on the independence of unobserved noise in the response variable on chosen actions can be strong in some applications. In looking at the work, I found the application to causal inference quite interesting and the latter assumption within that context a reasonable one. The authors are strongly encouraged to improve upon presentation and motivation if the paper is accepted.